# *NOG1* increases grain production in rice

Xing Huo[1], Shuang Wu[1], Zuofeng Zhu[1], Fengxia Liu[1], Yongcai Fu[1], Hongwei Cai [1], Xianyou Sun[1], Ping Gu[1], Daoxin Xie[2], Lubin Tan [1] & Chuanqing Sun[1]

During rice domestication and improvement, increasing grain yield to meet human needs was the primary objective. Rice grain yield is a quantitative trait determined by multiple genes, but the molecular basis for increased grain yield is still unclear. Here, we show that *NUMBER OF GRAINS 1* (*NOG1*), which encodes an enoyl-CoA hydratase/isomerase, increases the grain yield of rice by enhancing grain number per panicle without a negative effect on the number of panicles per plant or grain weight. *NOG1* can significantly increase the grain yield of commercial high-yield varieties: introduction of *NOG1* increases the grain yield by 25.8% in the *NOG1*-deficient rice cultivar Zhonghua 17, and overexpression of *NOG1* can further increase the grain yield by 19.5% in the *NOG1*-containing variety Teqing. Interestingly, *NOG1* plays a prominent role in increasing grain number, but does not change heading date or seed-setting rate. Our findings suggest that *NOG1* could be used to increase rice production.

---

[1] State Key Laboratory of Plant Physiology and Biochemistry, China Agricultural University, Beijing 100193, China. [2] Tsinghua-Peking Center for Life Sciences, School of Life Sciences, Tsinghua University, Beijing 100084, China. Correspondence and requests for materials should be addressed to L.T. (email: tlb9@cau.edu.cn) or to C.S. (email: suncq@cau.edu.cn)

Rice (*Oryza sativa* L.), one of the earliest domesticated crops, feeds over one-third of the population in the world as a staple food[1]. During the evolutionary course of cultivated rice from wild rice (*Oryza rufipogon* Griff.), marked morphological changes have occurred. In recent years, several key genes, which control the critical transitions of morphological traits such as seed shattering, prostrate growth habit, grain color, the spreading panicle, and awn development, have been isolated in rice[2–15]. The transition from less grain number per panicle to a greater number per panicle (Fig. 1a, b) is one of the most important events in rice domestication. After the domestication, further selection would occur for pursuing high-yield production, which could meet the needs for food as the global population increases. Grain yield of rice is a complicated trait and is directly determined by three main components, including the number of panicles per plant, the number of grains per panicle, and grain weight. In recent years, several genes controlling yield-related traits have been characterized[16–30]. However, the molecular mechanisms increasing the grain yield during rice improvement are still largely unknown.

Here, we report the identification of a gene, *NUMBER OF GRAINS 1* (*NOG1*), which regulates grain number and the yield of rice. We find that *NOG1* encodes an enoyl-CoA hydratase/isomerase (ECH), a key enzyme in fatty acid β-oxidation pathway, and upregulation of *NOG1* transcript levels significantly enhances grain number and yield without negative effects in the number of panicles, grain weight, seed-setting rate, and heading date. The identification of *NOG1* will not only enhance our understanding for the molecular basis of regulation of grain yield, but will also provide a favorable gene for breeding high-yield rice.

## Results

**Cloning of *NOG1*.** We identified an introgression line SIL176 with less grain number and low grain yield, which carried six introgressed segments from Dongxiang common wild rice (DXCWR, *O. rufipogon* Griff.) in a high-yielding *indica* variety Guichao 2 background[31] (Supplementary Fig. 1). Compared with Guichao 2, grain number per main panicle and grain yield per plant of SIL176 decreased by 39.6% and 44.3%, respectively (Fig. 1c–e). Phenotype observations found that primary branch number, grain number of primary branches, the secondary branch number, and grain number of secondary branches of SIL176 were significantly less than those of Guichao 2 (Supplementary Fig. 2a–d). However, no significant differences were observed in the number of panicles per plant, 1000-grain weight, seed-setting rate, and heading date between SIL176 and Guichao 2 (Supplementary Fig. 2e–h).

The F$_1$ plants from the cross between SIL176 and Guichao 2 displayed similar phenotypes of grain number per main panicle and grain yield per plant to Guichao 2 (Supplementary Fig. 2i, j). A genetic linkage analysis of 243 F$_2$ individuals derived from the cross between SIL176 and Guichao 2 showed that there is a major QTL, *NOG1*, which locates between SSR markers RM1183 and RM297 on chromosome 1, and explains 13% of phenotypic variance of grain number per main panicle and 12% of phenotypic variance of grain yield per plant. The allele from Guichao 2 at the locus could increase the grain number per main panicle and grain yield per plant (Supplementary Table 1).

To score the phenotype precisely and conveniently, we selected the number of grains per main panicle as the targeted trait for fine-mapping *NOG1*. Progeny test of 6230 F$_2$ individuals derived from the cross between Guichao 2 and SIL176 revealed that *NOG1* was delimited to a 4.3-kb region between the single-nucleotide polymorphism (SNP) marker 860CX5 and 870CX2. Annotation according to the MSU Rice Genome Annotation Project Database (http://rice.plantbiology.msu.edu/) showed that the 4.3-kb mapping region harbors the promoter (*cis*-regulatory region) and the first three exons of LOC_Os01g54860, and the last exon of LOC_Os01g54870 (Fig. 2a). Sequencing the 4.3-kb mapping region revealed that there was one 12-bp insertion–deletion (InDel) and 15 SNPs between Guichao 2 and SIL176 in the promoter region of LOC_Os01g54860 (Supplementary Fig. 3a

To verify the function of LOC_Os01g54860, we generated a complementary construct (pCPL), which contained a 6903-bp fragment, including the entire gene region and 2504-bp promoter

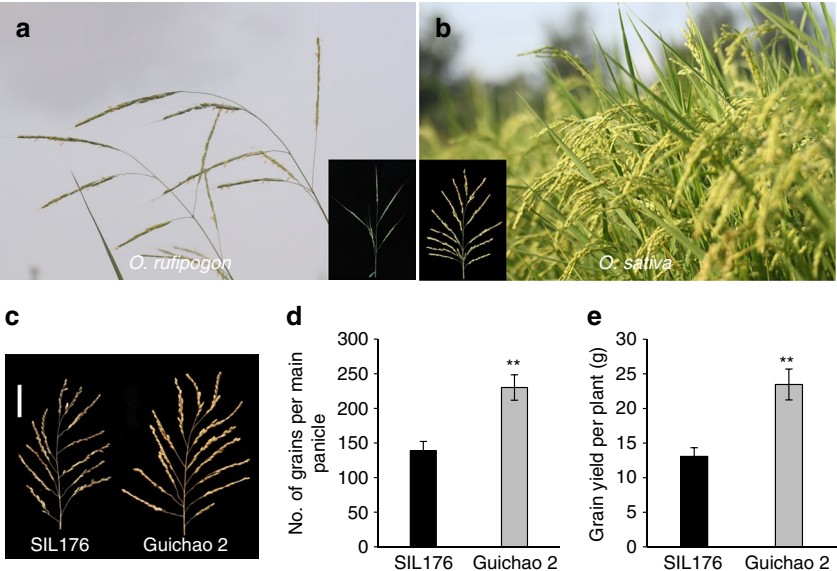

**Fig. 1** Comparison of a phenotype. **a**, **b** The transition of grain number from wild rice (*Oryza rufipogon*) to cultivated rice (*Oryza sativa*). A panicle is shown in the lower corner, respectively. **c** The main panicle of SIL176 and Guichao 2. Scale bars, 5 cm. **d**, **e** Comparison of the number of grains per main panicle and grain yield per plant between SIL176 and Guichao 2. Data are means, with error bars showing the standard error of mean (SEM), $n = 30$ (two-tailed Student's *t* test; **$P < 0.01$)

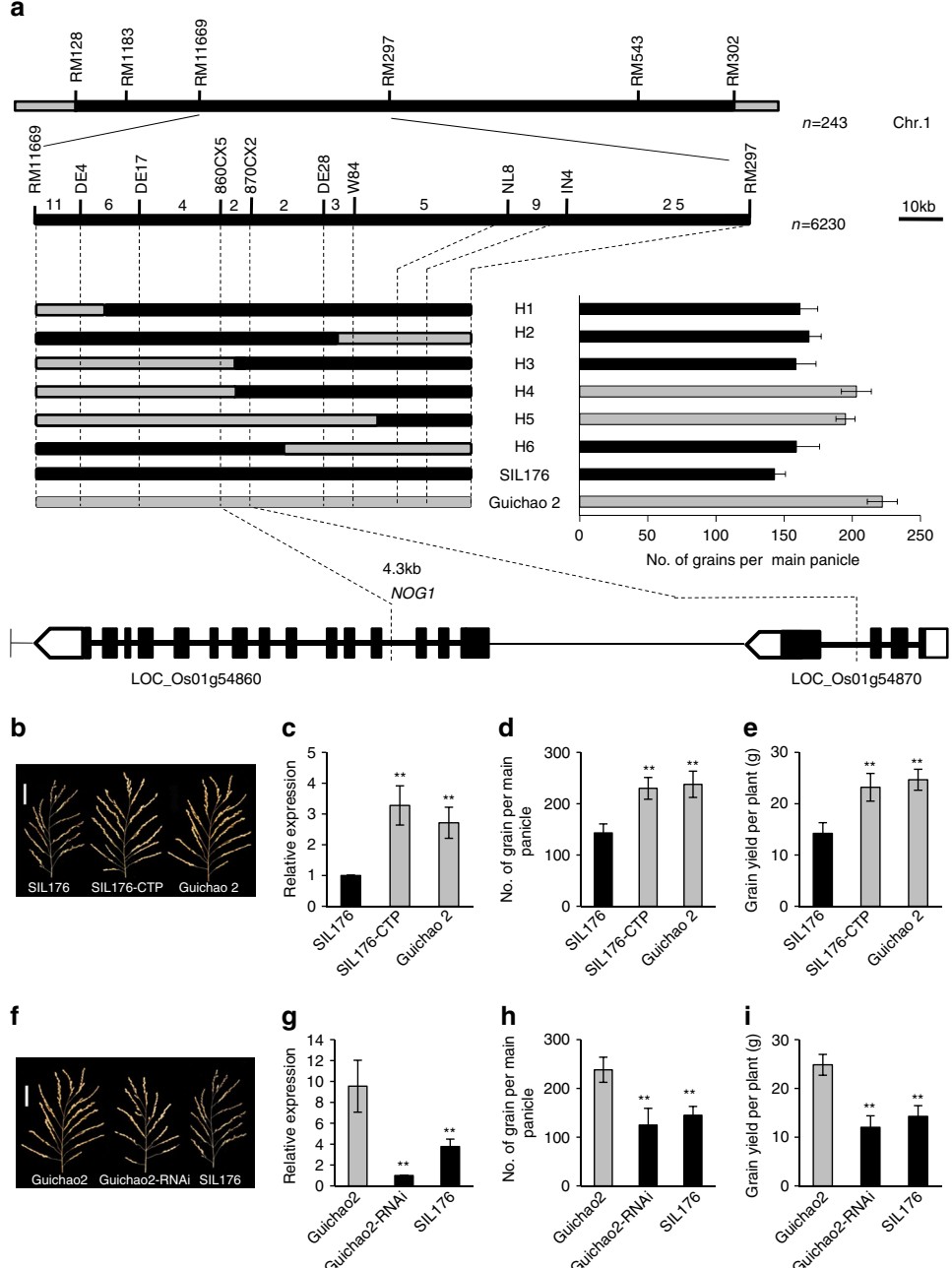

**Fig. 2** Mapping and cloning of *NOG1*. **a** The *NOG1* locus was detected on chromosome 1. Positional cloning narrowed the *NOG1* locus to a 4.3-kb region between 860CX5 and 870CX2. The numbers above the bars indicated the number of recombinants. LOC_Os01g54860 was the candidate gene. **b** The main panicle of SIL176 (control), complemented transgenic plants (SIL176-CTP), and Guichao 2. Scale bars, 5 cm. **c** Comparison of the relative expression levels of *NOG1* among SIL176, SIL176-CTP, and Guichao 2 (n = 4). **d**, **e** Comparison of grain number per main panicle and grain yield per plant among SIL176, SIL176-CTP, and Guichao 2 (n = 30). **f** The main panicle of Guichao 2 (control), RNAi transgenic plants (Guichao 2-RNAi), and SIL176. Scale bars, 5 cm. **g** Comparison of the relative expression levels of *NOG1* among Guichao 2, Guichao 2-RNAi, and SIL176 (n = 4). **h**, **i** Comparison of grain number per main panicle and grain yield per plant among Guichao 2, Guichao 2-RNAi, and SIL176 (n = 30). Data are means, with error bars showing SEM (two-tailed Student's *t* test; **P < 0.01)

region of LOC_Os01g54860 from Guichao 2, and introduced the pCPL construct into SIL176 using *Agrobacterium*-mediated transformation. Compared with the SIL176 (control), relative expression levels of LOC_Os01g54860 were elevated, and grain number per main panicle and grain yield per plant were increased in the complemented transgenic plants (SIL176-CTP) (Fig. 2b–e), but there were no changes in the number of panicles per plant, 1000-grain weight, seed-setting rate, and heading date (Supplementary Fig. 4a–d).

To further validate the function of LOC_Os01g54860, we transformed an RNA interference construct (pRNAi) into Guichao 2. We found that the LOC_Os01g54860 transcript levels reduced, and grain number per main panicle and grain yield per plant decreased in the RNAi transgenic plants (Guichao 2-RNAi) compared with control plants (Fig. 2f–i). We also found that there were no significant differences in the number of panicles per plant, 1000-grain weight, seed-setting rate, and heading date between transgenic plants and control (Supplementary

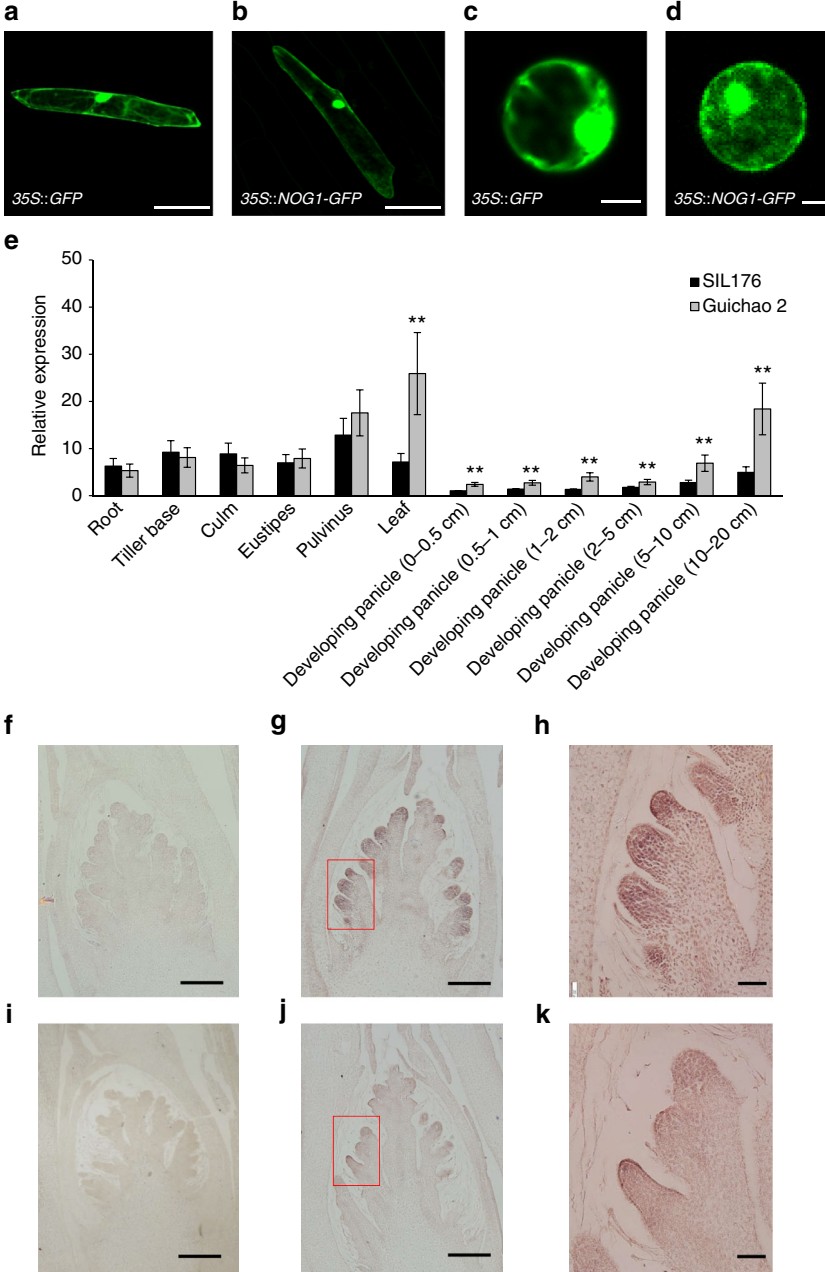

**Fig. 3** Transcriptional characterization of *NOG1*. **a–d** The subcellular localization of NOG1 protein. **a**, **b** The *p35S::GFP* and *p35S::NOG1-GFP* fusion gene were transiently expressed in onion inner epidermal cells. Scale bars, 100 μm. **c**, **d** Subcellular localization of the *p35S::GFP* and *p35S::NOG1-GFP* fusion gene in rice protoplast cells. Scale bars, 5 μm. The NOG1-GFP protein was localized in both the nucleus and the cytosol. **e** The expression pattern of *NOG1* in SIL176 and Guichao 2. Expression analysis of *NOG1* in root, tiller base, culm, eustipes, pulvinus, leaf, and young panicles. Data are means, with bars showing SEM, $n = 3$. (two-tailed Student's $t$ test; **$P < 0.01$). **f–h** Expression patterns analysis of *NOG1* revealed by messenger RNA (mRNA) in situ hybridization during panicle development in Guichao 2. **f** Sense probes as negative controls. Scale bars, 100 μm; **g** RNA in situ hybridization of *NOG1* during panicle development. Scale bars, 100 μm; **h** Enlarged from **g** marked by a red square. Scale bars, 20 μm. **i–k** Expression patterns analysis of *NOG1* revealed by mRNA in situ hybridization during panicle development in SIL176. **i** Sense probes as negative controls. Scale bars, 100 μm; **j** RNA in situ hybridization of *NOG1* during panicle development. Scale bars, 100 μm; **k** Enlarged from **j** marked by a red square. Scale bars, 20 μm

Fig. 4e–h). Taken together, these results demonstrated that LOC_Os01g54860 is *NOG1* that is essential for regulating the grain number per panicle and grain yield per plant in rice.

**Transcriptional characterization of *NOG1*.** Sequence analysis of 5′- and 3′-rapid amplification of complimentary DNA (cDNA) ends (RACE) and cDNA products indicated that the *NOG1* cDNA in Guichao 2 is 1479-bp long, with an open reading frame of 1167-bp, a 159-bp 5′-untranslated region, and a 153-bp

3′-untranslated region, and it encodes a 388-amino-acid protein (Supplementary Fig. 3b). We constructed a vector harboring *NOG1-GFP* (green fluorescence protein) fusion gene driven by a *CaMV35S* promoter, and then introduced a plasmid into onion epidermal cells and rice protoplast cells. The fluorescent signal showed that the NOG1 protein was localized in the nucleus and cytoplasm (Fig. 3a–d).

We investigated the tissue specificity of *NOG1* expression using real-time quantitative PCR (RT-qPCR). The results showed that

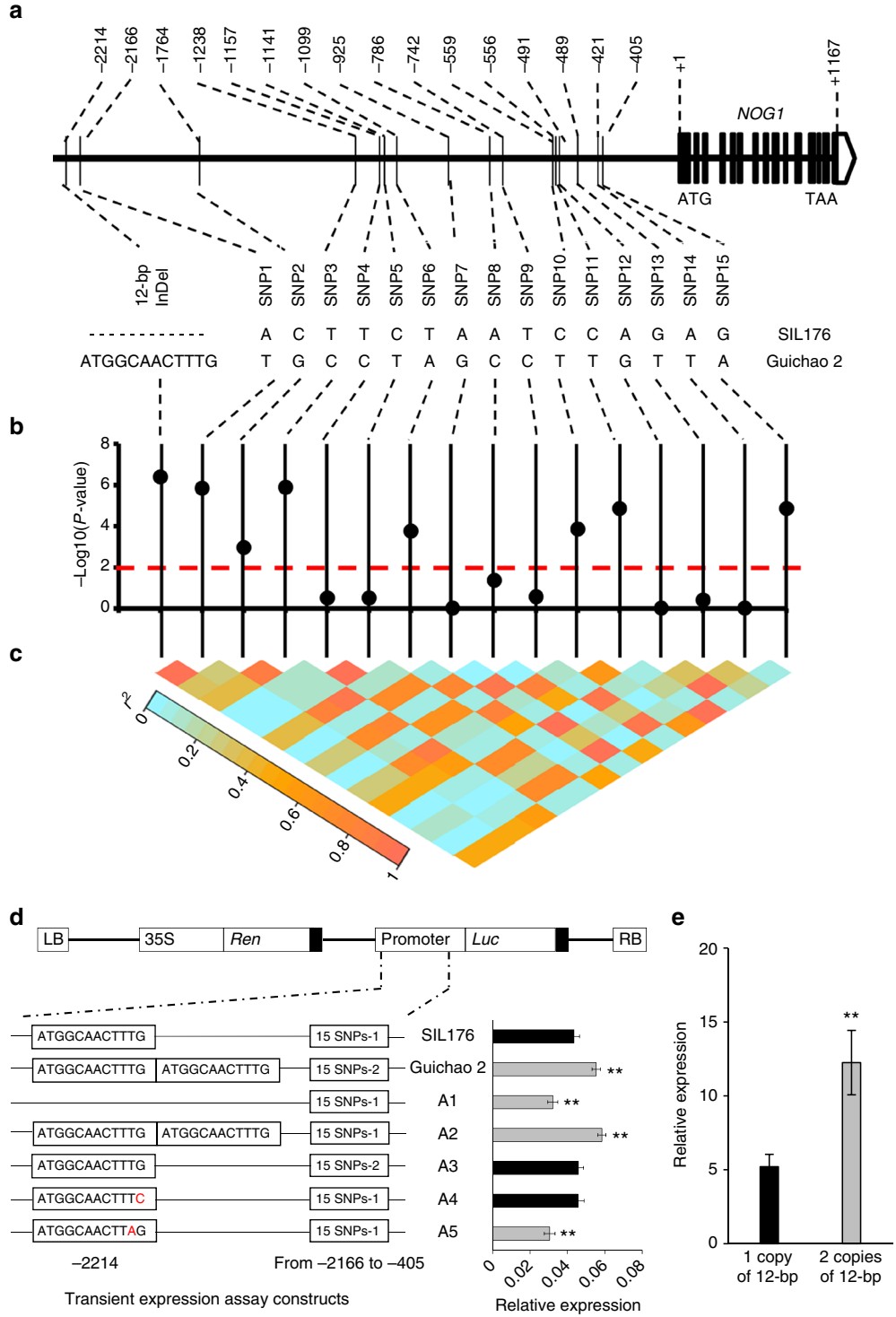

**Fig. 4** Association test and expression profile of *NOG1* variation. **a** Sequences comparison of a promoter region between SIL176 and Guichao 2.
**b** Association testing with grain number per main panicle of 16 variants in the 4.3-kb fine-mapping region. Black dots represent 16 variations. **c** Triangle matrix of pairwise linkage disequilibrium. **d** Effect of the 12-bp insertion and 15 SNPs in *NOG1* promoter by transient expression assays. Above, constructs of pGreenII 0800-*LUC* vector. Left, constructs with site-directed *NOG1* promoter. SIL176 and Guichao 2 define the constructs with the ~2500-bp promoter fragments from SIL176 and Guichao 2, respectively. The SIL176 promoter contained one copy of the 12-bp and the 15 SNPs-type 1 (ACTTCTAATCCAGAG). The Guichao 2 promoter contained two copies of the 12-bp and the 15 SNPs-type 2 (TGCCTAGCCTTGTTA). A1, A2, and A3 define constructs with site-directed mutations at the 12-bp InDel (positions −2214) and 15 SNPs (positions from −2166 to −405) in the SIL176 promoter fragment, respectively. A4 and A5 define constructs with site-directed mutations at TBOXATGAPB and DOFCOREZM TFBS on the 12-bp InDel in the SIL176 promoter fragment, respectively. Right, expression levels of firefly luciferase relative to Renilla luciferase (*n* = 10). **e** Comparison of *NOG1* gene expression levels between varieties containing one copy of 12-bp and varieties containing two copies of 12-bp (*n* = 20). Data are means, with error bars showing SEM (two-tailed Student's *t* test; **P* < 0.01)

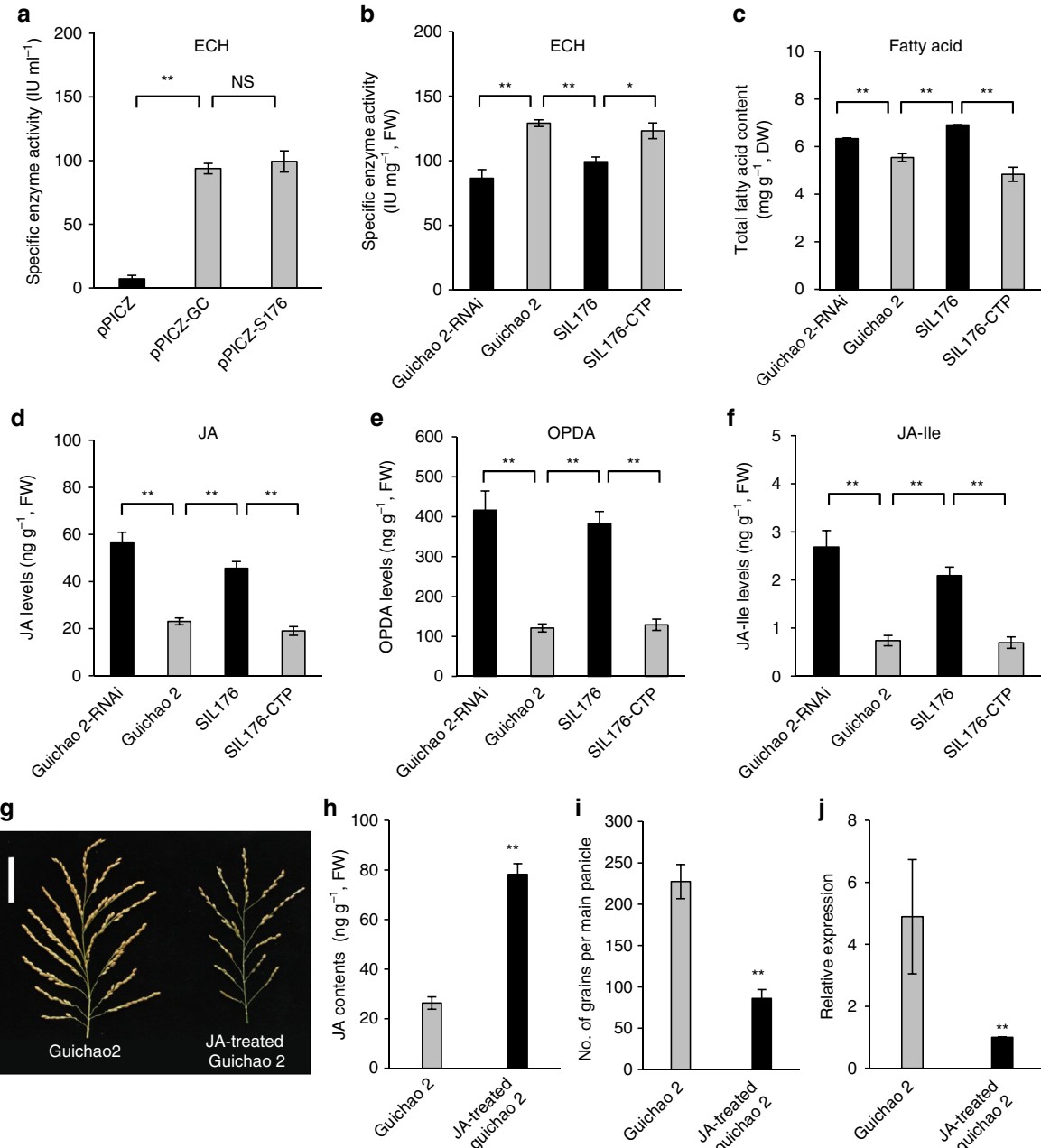

**Fig. 5** Functional analysis of *NOG1*. **a** ECH enzymatic activity assays of NOG1/nog1 and 6×His fusion protein by eukaryotic expression in *Pichia* yeast. The empty pPICZ-C vector plasmid was transfected as control (pPICZ), and pPICZ-GC and pPICZ-S176 were transfected by *NOG1* and *nog1*, respectively (*n* = 10). **b** Enzymatic activity assays of ECH in young panicles among Guichao 2, SIL176, and transgenic plants (*n* = 10). **c** Comparison of the total fatty acid content in young panicles among Guichao 2, SIL176, and transgenic plants (*n* = 3). **d–f** Comparison of JA, OPDA, and JA-Ile levels in young panicles among Guichao 2, SIL176, and transgenic plants (*n* = 10). **g** Main panicle of Guichao 2 and JA-treated Guichao 2. Scale bars, 5 cm. **h, i** Comparison of JA contents and the number of grains per main panicle between Guichao 2 and JA-treated Guichao 2 (*n* = 10). **j** Comparison of *NOG1* expression levels between Guichao 2 and JA-treated Guichao 2 (*n* = 3). Data are means, with error bars showing SEM (two-tailed Student's *t* test; **\*\***P < 0.01; \*P < 0.05; NS, P > 0.05)

*NOG1* expression was detected in various rice organs, including root, tiller base, culm, eustipes, pulvinus, leaf, and young panicles, and *NOG1* relative expression levels in the leaf and young panicles of Guichao 2 were significantly higher than that of SIL176 (Fig. 3e). At the same time, the expression levels of F$_1$ plants in young panicles were similar with Guichao 2, consistent with a phenotype (Supplementary Fig. 2k). RNA in situ hybridization signals of *NOG1* in young panicles indicated that *NOG1* is expressed in the primordia of branches, and hybridization signals in Guichao 2 were stronger than that in SIL176

(Fig. 3f–k). The high levels of *NOG1* expression in developing panicles are consistent with its role in controlling the grain number in rice.

**A 12-bp insertion enhances *NOG1* expression and grain number.** Sequence analysis showed that there was one 12-bp InDel (ATGGCAACTTTG) and 15 SNP variations in the promoter region of *NOG1* between SIL176 and Guichao 2 (Fig. 4a). Interestingly, there are two closely connected copies of the 12-bp, which locates the 2214-bp upstream of the translation start site in

Guichao 2, but there is only one copy of the 12-bp in SIL176. Further analysis of the promoter of *NOG1* using a database of Plant Cis-acting Regulatory DNA Elements (PLACE, www.dna. affrc.go.jp/PLACE/signalup.html), showed that there are two transcription factor-binding sites (TFBS) within the 12-bp InDel, one is DOFCOREZM (−) AAAG (DOF TFBS) and another is TBOXATGAPB (+) ACTTTG.

To determine the relationships between the mutations in the *NOG1* promoter region and the grain number, we sequenced the 4.3-kb fine-mapping region of 158 cultivars, including 84 *indica* and 74 *japonica* varieties. An association test between grain number per main panicle and sequence variations in the promoter region showed that the strongest signal was present at the 12-bp InDel and 7 SNPs (Fisher's exact test, $P < 0.01$) (Fig. 4b, c).

To verify whether the 12-bp InDel and 15 SNPs in the *NOG1* promoter region affected the *NOG1* gene expression, we generated constructs by installing mutant promoter fragments into the binary vector pGreenII 0800-*LUC*, and introduced the constructs into rice protoplasts for transient expression assays. Compared with the transient relative activity of the SIL176 promoter, the relative activities of mutant promoter fragment with 15 SNP variations but without 12-bp InDel (A3 construct) did not change. It suggested that 15 SNPs have no impact on the expression of *NOG1* gene. Furthermore, deletion of the 12-bp (A1 construct) and mutation of DOF TFBS in 12-bp (A5 construct) significantly reduced the transcriptional activity of the SIL176 promoter. Adding one copy of 12-bp (A2 construct) enhanced the transcriptional activity of the SIL176 promoter, as well as the Guichao 2 promoter (Fig. 4d).

To further verify whether the 12-bp InDel was responsible for the regulation of *NOG1* gene expression, 20 cultivars containing two copies of 12-bp and 20 cultivars containing one copy of 12-bp were selected to investigate their *NOG1* gene expression. The results showed that the expression levels of *NOG1* gene in these varieties containing two copies of 12-bp were obviously higher than that in varieties containing one copy of 12-bp (Fig. 4e). The results suggest that the 12-bp insertion with DOF TFBS in the *NOG1* promoter region might be responsible for the *NOG1* expression levels among rice accessions.

To ascertain whether expression levels of *NOG1* affect the grain number, we generated the overexpression construct harboring *nog1* cDNA from SIL176, driven by the maize ubiquitin promoter. The construct was introduced into an elite *japonica* rice variety Zhonghua 17 that harbors the *nog1* cDNA identical to SIL176. Compared with Zhonghua 17 as a control, *NOG1* expression was significantly enhanced, and grain number per panicle and grain yield per plant were significantly increased in the overexpression transgenic plants (Supplementary Fig. 5). The results indicated that upregulation of the *NOG1* expression levels could increase the grain number per panicle and grain yield per plant in rice. Taken together, the 12-bp insertion in the *NOG1* promoter region upregulates the *NOG1* expression, resulting in the increase of grain number and grain yield in rice.

**NOG1 expression affects the ECH levels**. A protein-BLAST (BLASTp) at NCBI online and gene identification by MSU-RGAP revealed that *NOG1* encodes an ECH that catalyzes a reversible syn-hydration of *trans*-2,3-enoyl-CoA to the corresponding L-3-hydroxyacyl-CoA[32]. Compared with the *nog1* cDNA of SIL176, *NOG1* cDNA of Guichao 2 has a 3-bp deletion on the 14th exon, resulting in an amino-acid deletion after being translated (Supplementary Fig. 3c, d).

To verify whether the NOG1/nog1 protein has ECH activity, the NOG1/nog1 protein was encoded by *NOG1/nog1*-coding sequence from Guichao 2/SIL176, driven by the *AOX1* promoter, which was eukaryotic expressed in yeast *Pichia pastoris*. We assayed the enzyme activity of eukaryotic expression products and found that the equivalent amount of NOG1 and nog1 protein generated a similar ECH enzyme activity (Fig. 5a). These results indicated that the NOG1 protein has ECH activity, and two proteins of NOG1 and nog1 have equal enzymatic activity; thus, the 3-bp InDel in the *NOG1*-coding sequence did not affect the NOG1 protein function, which was consistent with the results that the 3-bp deletion was out of the 4.3-kb fine-mapping region.

Further investigation of the ECH enzyme activity in young panicles showed that ECH enzyme levels both in Guichao 2 and the complemented transgenic plants (SIL176-CTP) were higher than that in SIL176, while ECH enzyme levels in the RNAi transgenic plants (Guichao 2-RNAi) were lower than that in the control plants (Guichao 2) (Fig. 5b). These results suggested that the *NOG1* expression affected the ECH enzyme levels, and upregulation of the *NOG1* expression increased the ECH enzyme levels in plants.

**NOG1 expression regulates fatty acid β-oxidation**. A previous study revealed that ECH is a key enzyme in the second step of fatty acid β-oxidation pathway, and leads to component accumulation of the long-chain fatty acids[32]. In order to determine whether a change of ECH enzyme levels affected fatty acid β-oxidation system, we measured fatty acids content and their composition in young panicles, using gas-liquid chromatography analysis, and found that the total fatty acids content and linolenic acid (C18:3) in Guichao 2, and the complemented transgenic plants were lower than those in the SIL176 and the RNAi transgenic plants, respectively (Fig. 5c; Supplementary Table 2).

**NOG1 decreases plant hormone jasmonic acid levels**. The β-oxidation pathway participates in not only the catabolism of fatty acids, but also in the metabolism of hormones and amino acids[33]. We measured the endogenous jasmonic acid (JA), 3-indoleacetic acid (IAA), and salicylic acid (SA) levels in young panicles by high-performance liquidchromatography–mass spectrometry (HPLC-MS/MS) method and found that Guichao 2 and the complemented transgenic plants showed lower endogenous JA levels, compared with SIL176 and the RNAi transgenic plants (Fig. 5d), but there was no difference in IAA and SA levels among Guichao 2, SIL176, and transgenic plants (Supplementary Fig. 6).

Previous studies revealed that C18:3 is the synthetic precursor of JA, and peroxisomal β-oxidation is required to convert 12-oxo-phytodienoicacid (OPDA) into JA in the last few steps of JA synthesis[34, 35]. We examined the contents of jasmonates including OPDA and JA-isoleucine (JA-Ile) in young panicles, and compared with SIL176 and the RNAi transgenic plants, Guichao 2 and the complemented transgenic plants showed lower OPDA and JA-Ile levels, consistent with JA levels (Fig. 5e, f). Furthermore, compared with SIL176, the expression levels of *OsDAD1*, *OsLOX*, *OsAOS2*, *OsAOC*, and *OsOPR7* associated with JA biosynthesis pathway were downregulated in Guichao 2 (Supplementary Fig. 7a–e).

JA is one of the most important signal molecules regulating the development of plants, and higher JA levels might inhibit the development of rice spikelet and decrease grain number[36, 37]. To further confirm whether the JA affects the grain number, we treated Guichao 2 with MeJA at the panicles development stage. After an excess of 100 mM MeJA treatment, the JA contents were increased in Guichao 2. In addition, the grain number per panicle of JA-treated Guichao 2 was decreased. Further examination showed that the *NOG1* gene expression in Guichao 2 was downregulated by JA treatment (Fig. 5g–j). These results

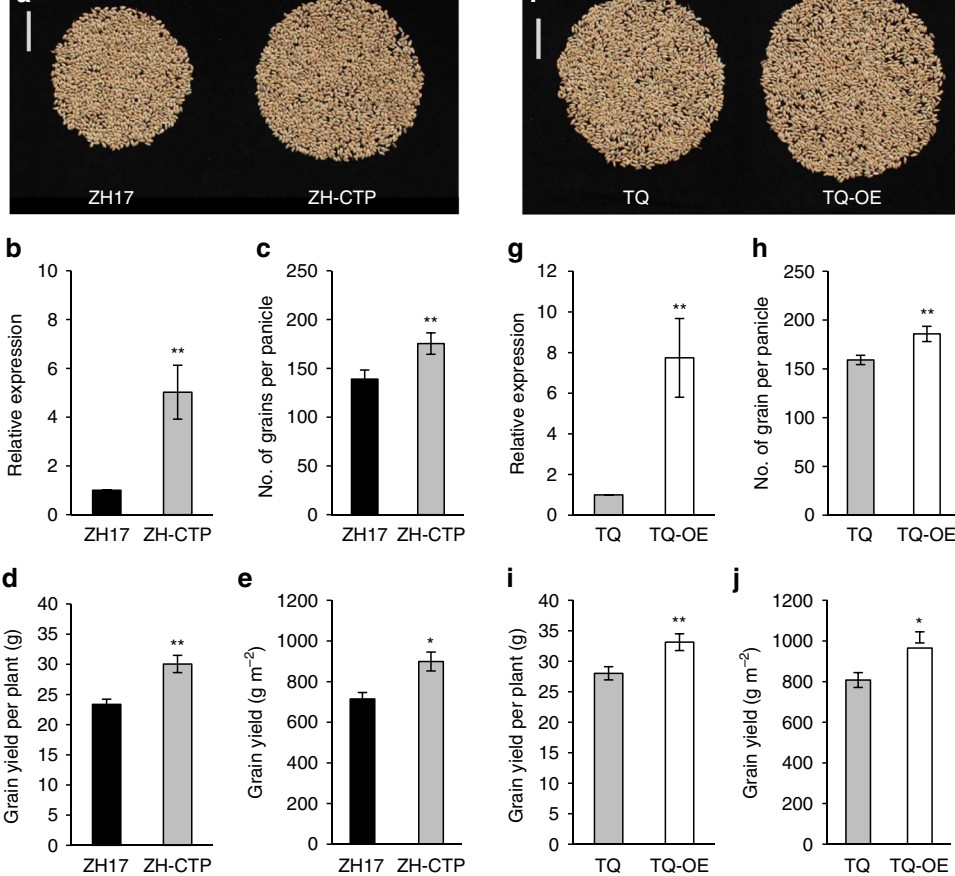

**Fig. 6** *NOG1* increases grain production in cultivars. **a** Grains from one plant of Zhonghua 17 (ZH17) and complemented transgenic plants (ZH-CTP). Scale bars, 5 cm. **b** Comparison of *NOG1* expression levels between ZH17 and ZH-CTP ($n = 4$). **c**, **d** Comparison of the number of grains per panicle and grain yield per plant between ZH17 and ZH-CTP ($n = 30$). **e** Comparison of grain yield in the field between ZH17 and ZH-CTP ($n = 5$). **f** Grains from one plant of Teqing (TQ) and *NOG1* overexpression transgenic plants (TQ-OE). Scale bars, 5 cm. **g** Comparison of *NOG1* expression levels between TQ and TQ-OE ($n = 4$). **h**, **i** Comparison of the number of grains per panicle and grain yield per plant between TQ and TQ-OE ($n = 30$). **j** Comparison of grain yield in the field between TQ and TQ-OE ($n = 5$). Data are means, with error bars showing SEM (two-tailed Student's *t* test; **$P < 0.01$; *$P < 0.05$)

demonstrated that the JA treatment downregulated *NOG1* gene expression and decreased the grain number in rice.

**Origin of the *NOG1* gene**. In order to explore the origin and distribution of the 12-bp insertion in rice, we sequenced an ~2.5-kb genomic fragment of *NOG1* promoter in 158 varieties of cultivated rice and 75 accessions of wild rice. The results showed that 75 accessions of wild rice all harbor one copy of the 12-bp functional site identical to SIL176. The single copy of the 12-bp fixed in wild rice might be adaptive in the wild. However, among 158 varieties of cultivated rice, 61 varieties (48 *indica* and 13 *japonica* varieties) harbor two copies of the 12-bp functional sites identical to Guichao 2, and 97 varieties (36 *indica* and 61 *japonica* varieties) harbor one copy of 12-bp (Supplementary Tables 3, 4), indicating that the mutation of the 12-bp insertion in *NOG1* of cultivated rice might occur after the transition from wild rice to cultivated rice. We also found that the proportion harboring 12-bp insertion in *indica* varieties (57.1%) was much higher than that in *japonica* varieties (17.6%), suggesting that the 12-bp variation in *NOG1* promoter region might have originated in the *indica* cultivars and was later introduced into *japonica* cultivars.

**NOG1 enhances the grain yield of rice**. As 61.4% of cultivars did not harbor the beneficial 12-bp insertion, we selected the commercial high-yielding *japonica* cultivar Zhonghua 17, which

contains one copy of 12-bp, and is widely used in production. Compared with Zhonghua 17 as a control, the *NOG1*-complemented transgenic plants exhibited an obvious increase in the *NOG1* expression levels, grain number per panicle, and grain yield per plant, while there was no significant difference in other traits, including the number of panicles per plant, 1000-grain weight, seed-setting rate, and heading date between transgenic plants and control. Field estimation of grain yield showed that the *NOG1*-complemented transgenic plants increased the grain yield by 25.8% compared with the control (Fig. 6a–e; Supplementary Fig. 8a–d).

To further explore the application potential of *NOG1* in rice breeding, we introduced an overexpression construct harboring *NOG1*-coding sequence into an elite commercial *indica* cultivar Teqing that harbors two copies of 12-bp. Compared with Teqing as a control, the expression levels of *NOG1* in the transgenic plants were significantly elevated. Consistent with the high expression of *NOG1*, grain number per panicle and grain yield per plant were significantly increased in the overexpression transgenic plants, while changes in other traits were not observed. Field estimation of grain yield showed that the *NOG1* overexpression transgenic plants increased the grain yield by 19.5% compared with the control (Fig. 6f–j; Supplementary Fig. 8e–h). These results indicated that upregulating *NOG1* expression can further increase the grain yield in *NOG1*-containing high-yield varieties.

## Discussion

Here, we show that the *NOG1* gene can increase the grain number and yield. *NOG1* encodes an enoyl-CoA hydratase/isomerase, and a 12-bp insertion in the promoter region upregulates the *NOG1* gene expression, elevates the levels of the enoyl-CoA hydratase, decreases the total fatty acids and linolenic acid contents, downregulates endogenous JA levels, and enhances the grain number to increase grain yield in rice. A field test showed that introduction of *NOG1* increases the grain yield by 25.8% in the *NOG1*-deficient rice cultivar Zhonghua 17, and overexpression of *NOG1* can further increase the grain yield by 19.5% in the *NOG1*-containing variety Teqing. *NOG1* can significantly increase the grain yield of commercial high-yield varieties, which serves as a favorable gene to increase rice production.

During the process of rice domestication from wild rice to cultivated rice, not only the morphological characters changed greatly, but also the grain yield of cultivated rice was improved significantly compared with its progenitors. Previous studies revealed that the transitions of morphological traits, such as seed shattering, prostrate growth habit, grain color, the spreading panicle, and awn development, are controlled by one or a few of the genes, respectively[2–15], and the differences of grain yield between wild and cultivated rice are controlled by a lot of QTLs[38, 39]. However, only a few genes that controlled yield improvement have been identified. In this study, we identify a key gene *NOG1* that increases the grain number of cultivated rice. Sequence analysis showed that some cultivars, both of *indica* and *japonica* varieties still harbor one copy of 12-bp, similar to wild rice. The introduction of *NOG1* increases the grain yield by ~25% in the *NOG1*-deficient variety, suggesting that *NOG1* is a key gene regulating the improvements of yield. Identification of *NOG1* provides not only molecular evidences for yield improvements, but also a favorable gene to increase rice production.

A change in the protein product caused by sequence variation on the gene-coding region is the most important molecular basis of allelic variation. However, changes of gene expression levels caused by the sequence variation on transcriptional regulatory region were also an important genetic basis of trait variation. Typical examples are *OsLG1* in rice, *tb1* in maize, and *fw2.2* in tomato[10, 40, 41], and the phenotypic changes are due to the sequence variation of their promoter regions. In this study, we find that a 12-bp repetition containing DOF TFBS in the *NOG1* promoter region results in an increase of the *NOG1* expression levels, leading to increasing the grain number. DOF protein is one kind of plant-specific transcription factor that plays an important role in a variety of physiological processes, including carbon metabolism, and recognizes the AAAG motif as a core-element sequence. Previous studies revealed that two concatenate-repeated AAAG motifs showed about a two-fold higher affinity for DNA than a probe containing one motif[42]. Overall, we speculate that the repetition of TFBS increases gene expression, resulting in increasing the grain number and enhancing grain yield. We can increase the grain yield by upregulating the *NOG1* gene expression levels in rice breeding.

The number of panicles per plant, the number of grains per panicle, and grain weight are the three important elements, and are also the important goals of breeding improvement. During the improvement of rice, changes in the target traits by genetic modification often lead to problematic agronomic traits associations because of pleiotropic gene effects. For example, *IPA1* increases the grain number and grain weight but reduces panicle number[17]; *Ghd7* increases the grain number per panicle but delays heading date[20]; and *GW2* increases the grain size and the number of panicles per plant but delays heading date and reduces the grain number per panicle[24]. It is essential to rice breeding that we identify novel genes improving yield traits without any trade-offs for other agronomic or yield-related traits. The *NOG1* gene identified in this study increases the grain number without adverse effects on the number of panicles, grain weight, seed-setting rate, and heading date. Homology analysis showed that the orthologs of *NOG1* exist in various plants such as *Sorghum bicolor*, *Zea mays*, *Setaria italica*, *Triticum aestivum*, and so on (Supplementary Fig. 9). *NOG1* homologous genes may play a similar function in different crops; thus, it could be explored for general grain crop improvement. We believe that *NOG1* is a favorable gene for grain yield improvement, and the utilization of *NOG1* is very helpful for breeding high-yield varieties of grain crops.

## Methods

**Plant materials.** The introgression line, SIL176, was selected from ILs (BC$_4$F$_4$), Dongxiang common wild rice was used as a donor, and Guichao 2 was used as a recipient. Other materials used in this study are listed in Supplementary Tables 3, 4, including 84 *indica* (*O. sativa* L. ssp. *indica*) varieties, 74 *japonica* (*O. sativa* L. ssp. *japonica*) varieties, and 75 accessions of wild rice.

**Primers.** The primers used in this study are listed in Supplementary Table 5.

**Generation of constructs and transformation of rice.** The 6903-bp fragment including the entire gene region and 2661-bp promoter region of *NOG1* gene was amplified from Guichao 2 DNA, and was inserted into the binary vector pCAM-BIA1300 to form the complementary construction pCPL. The RNAi construct pRNAi contained an inverted repeat sequence harboring the 361-bp *NOG1* cDNA fragment. Two overexpression constructs pOE-GC and pOE-S176 harboring *NOG1* cDNA from Guichao 2 and *nog1* cDNA from SIL176, respectively, were driven by the maize ubiquitin promoter into the binary vector pCAMBIA1301. Depending on the different experimental purposes, these constructs were introduced into *Agrobacterium tumefaciens* strain EHA105 and subsequently transformed into SIL176, Guichao 2, Teqing, and Zhonghua 17 respectively, by *Agrobacterium*-mediated transformation. The phenotypes were detected in T$_2$ generation of transgenic plants. The testing of grain yield in the field was transplanted to 16.7 cm multiplied by 20 cm.

**RNA preparation and reverse transcription PCR.** The total RNAs were extracted from various tissues using an RNeasy Plant Mini Kit (Qiagen). First-strand cDNA was synthesized from about 2 μg of total RNAs in 20-μl volumes using Oligo (dT)$_{18}$ primer (Takara) and M-MLV reverse transcriptase (Promega).

**RACE and real-time quantitative PCR.** RACE (5'RACE and 3'RACE) were performed according to the manufacturer's instructions of 5'-Full RACE Kit and 3'-Full RACE Core Set (Takara), respectively. RT-qPCR was done on the CFX96 Real-Time System (Bio-Rad). Diluted cDNA was amplified using the SYBR Green Master Mix (Applied Biosystems). Each set of experiments was performed on at least three plants from each line and with at least three technical replications. The rice ubiquitin gene was used as the internal control, and the measurements were obtained using the relative quantification method[43].

**Subcellular localization.** We constructed a *p35S::NOG1-GFP* vector containing *NOG1*-coding sequence fused with the coding sequence of *GFP* in a frame driven by a *CaMV35S* promoter, and then introduced the plasmid into onion epidermal cells and rice protoplast. The bombarded tissues were examined with a Nikon C1 confocal laser microscope.

**RNA in situ hybridization.** The freshly young panicles of Guichao 2 and SIL176 were collected and fixed in FAA solution (50 ml of ethanol, 5 ml of acetic acid, 10 ml of 37% formaldehyde, and 35 ml of DEPC-H$_2$O) at 4 °C overnight, dehydration by ethanol from 50 to 100%, infiltration by xylene from 50 to 100%, and embedded in paraffin (Paraplast Plus, Fisher Scientific). The tissues were sliced into about 8–10-μm sections with a microtome (Leica RM2145), and then, the sections were mounted on RNase-free glass slides. The 192-bp 5'-region of *NOG1* FL-cDNA was amplified as the template, and digoxigenin-labeled sense and antisense RNA probes were prepared using a DIG Northern Starter Kit (Roche) following the manufacturer's instruction. RNA in situ hybridization with probes was performed on sections. Slides were observed using a microscope (Leica DMR) and photographed with a microcolor CCD camera (Apogee Instruments)[44].

**Transient expression assays of promoter activity.** The ~2.5-kb promoter fragments upstream of the *NOG1* translation start site were amplified from Guichao 2 and SIL176, respectively, and fragments with site-directed mutations at the 12-bp InDel and 15 SNPs were designed by an overlap extension of PCR. All of the

fragments were inserted into pGreenII 0800-*LUC*. The rice protoplast was isolated from rice leaves of 7–10 days after seeding under dark conditions. Each of the *NOG1* promoter–*LUC* gene fusion constructs was used for transient transformation into rice protoplasts. Activities of *LUC* to *REN* luciferase were measured using the Dual-Luciferase Reporter Assay System (Promega)[45, 46].

**Enzyme activity assays of ECH**. The coding sequence from Guichao 2 and SIL176 was inserted into eukaryotic expression vector pPICZ-C, respectively, and then two constructs and pPICZ-C empty vector plasmid as control were introduced into the yeast *P. pastoris* by electroporation. The ECH enzyme activity was detected by following the hydration of the double-bond band. The decrease in absorbance was measured at 263 nm, 28 °C in an assay mixture contains $0.2 \ mol \ l^{-1}$ potassium phosphate, $0.2 \ mg \ ml^{-1}$ bovine serum albumin, and $30 \ \mu mol \ l^{-1}$ crotonyl-CoA (Sigma) at a pH of 8.0. The extinction coefficient was using $6700 \ M^{-1} \ cm^{-1}$ as the calculated rate[47].

**Measurements of fatty acid composition**. The freshly young panicles of Guichao 2, SIL176, and transgenic plants were collected and ground in liquid nitrogen. After vacuum freeze-drying process, the total fatty acid content and long-chain fatty acids compositions of samples were measured by gas–liquid chromatography analysis[48].

**Measurements of endogenous hormone levels**. The young panicles of parental and transgenic plants were ground into powder in liquid nitrogen. Extraction and quantitative analysis of endogenous hormones from each sample (50 mg of a fresh plant tissue) was determined by high-performance liquid chromatography–mass spectrometry (HPLC–MS/MS) method[49].

**MeJA treatments**. At 90 days after planting, before panicles initiation, Guichao 2 was subjected to 30 days until the end of panicles development, by 100 mM MeJA (95%, Sigma-Aldrich) solution.

**Data analysis**. QTL analysis was detected by the Map Manager QTX[50]. Association analysis was performed with Fisher's exact test, as Fisher's exact test can be powerful for qualitative traits such as grain number per panicle[51]. Multiple sequences were aligned using ClustalX[52]. A phylogenetic tree was constructed by the MEGA 5[53].

**Data availability**. The authors declare that the data supporting the findings of this study are available within the paper and the Supplementary Information, or are available from the corresponding author upon request. Sequence data were deposited in the GenBank under the following accession numbers: MF687920 (*NOG1* cDNA from Guichao 2) and MF687921 (DXCWR).

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

## Acknowledgements

We thank the International Rice Research Institute, Chinese Rice Research Institute, Institute of Crop Sciences of Chinese Academy of Agricultural Sciences, Guangxi Academy of Agricultural Sciences, and Guangdong Academy of Agricultural Sciences for providing the wild rice and cultivated rice samples. This research was supported by the National Natural Science Foundation of China (Grants 91535301), the National Key R&D Program for Crop Breeding (2016YFD0100301), and Special Fund for Agro-Scientific Research in the Public Interest (201403002-1).

## Author contributions

C.S. designed and supervised this study. X.H. conducted map-based cloning, genetic transformation, gene expression analysis, gene functional analysis, and evolutionary analysis. S.W. constructed the introgression lines and phenotypic data. X.S. and Y.F. maintained plant materials and performed the field management. L.T., F.L., Z.Z., H.C. and P.G. conducted the collection of rice germplasm and phenotypic data. C.S., X.H. and D.X. analyzed the data and wrote the article.

## Additional information

**Competing interests:** The authors declare no competing financial interests.

