## [Peer Review File · Nature Communications]

Reviewers' comments:

Reviewer #1 (Remarks to the Author):

The manuscript by Huo et al. investigates genetic determinants of grain production in rice. Using a combination of genetic linkage analysis, fine-mapping and reverse genetics approaches, authors of this study identified and characterized the role of NOG1 (NUMBER OF GRAINS 1) in controlling rice grain yield and most significantly the number of grains per panicle. The NOG1 allele of the elite indica variety of Guichao 2 is associated with high grain number per panicle, while in the introgression line SIL176 for the Guichao 2 background, and which notably lacks a 12-bp insertion in the upstream promoting region of the introgressed NOG1 allele, lower expression of NOG1 associates with reduced grain number per panicle. Consistent with the importance of NOG1 expression levels for grain yield, RNAi suppression of NOG1 in Guichao 2 reduce grain number per panicle and Guichao 2 NOG1 expression under its native promoter in the SIL176 line elevates grain yield. Authors further show that the 12-bp InDel upstream of Guichao 2 NOG1 overlaps with the most intense association signal for grain yield in a fine-mapping linkage analysis of 158 cultivars. Promotor assays studies additionally support that transcription factor binding sites within this 12-bp are responsible for the higher expression levels of Guichao 2 NOG1 compared to in SIL176. NOG1 encodes an enoyl-CoA hydratase/isomerase acting within the fatty acid β -oxidation pathway and results for Guichao 2 into higher total levels of fatty acids during grain maturation.

Forward and reverse genetics studies conducted in the first part of the article are very thorough and provide solid evidences for the importance NOG1 for grain production. The identification of NOG1 is a new and very interesting result for the field of crop improvement. However, the second part of the study that deals with the role of the jasmonate pathway for the NOG1-dependent control of grain yield, is in my view not as solid as the first.

1. The authors indeed detect that JA levels are higher in SIL176 than in Guichao 2 and show that JA application inhibits grain yield. Nevertheless, it remains elusive whether attenuated JA signaling in Guichao2 is the main cause for the NOG1-dependent increases of grain in Guichao 2 is not clear to me. I was notably wondering whether the authors analyzed JA-Ile, the bioactive form in the JA pathway, as well as downstream genes markers for JA signaling to ascertain of an attenuated JA signaling in Guichao 2.

2. β -oxidation is not only important for fatty acid turnover, as also demonstrated in this study by measurements of total fatty acid contents, but also in the last steps of JA synthesis. Peroxisomal β -oxidation is required to convert OPDA into JA. NOG1 may act more specifically at this stage of the JA pathway. Hence, it would have been interesting to examine OPDA levels to control for

a metabolic disruption at stage of the JA pathway. More generally, it seems to me that comparing JA biosynthetic genes expression between Guichao 2 and SIL176 would have been instrumental to better assess modulations of JA signaling between these two lines.

3. If C18:3 is limiting in Guichao 2 for JA synthesis, does a C18:3 increased JA levels in this line and consequently reduced grain yield to levels observed during JA treatment?

4. L99 seems to be a too strong statement. NOG1 is certainly involved, as demonstrated in this study, grain yield; but likely not “the” unique gene influential in this process.

Reviewer #2 (Remarks to the Author):

I have completed review of the manuscript by Huo et al. This manuscript identifies a novel grain yield gene called NOG1 and elegantly functionally characterizes it as well as its prevalence in Asian rice varieties and in wild rice.

This finding should be extremely useful to rice breeders and to all plant genetics, that major effect genes underlying traits of interest are still out there to be discovered. I found little to revise in this manuscript other than grammar. The flow is good and the approach is clear and sound.

If the editor agrees, it would be opportune to broaden discussion of if there are orthologs in other grains that should be explored for general grain crop improvement, and it would be useful to add a sentence about the earliest evidence of indica being introgressed into japonica (much was the other way around). There are many recent papers on this topic.

It would also be useful to add a sentence to the discussion about the role of jasmonic acid response in rice production and in domestication, if there are other changes that have been documented involving this pathway.

The abstract could be strengthened to provide more detail about what was found that relates to domestication.

The supplementary tables and figures were also excellent and sufficient, and the statistics and significance tests were appropriate and correct. I only found a few places that were unclear.

Overall it is a very strong study and I highly recommend Nature Communications publish it.

Line 20-21: unclear what genetically applied means. Are the authors trying to say genes controlling these traits haven't been pointed to as historic domestication genes?

Line 21: change to "in rice breeding programs"

Lines 30-31: If "Our findings offer new insights into the transition from a lower grain

Line 31: yield to a higher grain yield in rice domestication" then you have to include earlier in the abstract something about evidence that this gene was under selection during the domestication process.

Line 47: "characterized 16-30, only a" should be "characterized 16-30. However, only a"

Lines 49-50: again, be more specific about what applications you are referring to, and what a

domestication-related gene is. This really requires definition, especially as the community has defined, delimited, and debated domestication genes from diversification genes, and you are introducing a word that embraces both. It's a good word to use but just requires definition, such as giving examples of hallmark domestication genes and the signatures of selection on them from DNA evidence and/or archaeological evidence that let's us understand that these caused the changes observed nearly 9000 years ago.

Line 50: pluralize program

Lines 62: pluralize panicle

Line 73-74: we selected number of grains per main panicle as "the" targeted trait for fine-mapping NOG1.

Line 76: change "combining with phenotype" to "using the F3 generation averaged phenotypes" (if they are averaged)

Line 79: harbors

Line 84: change 'harbored' to 'contained'.

Line 87: levels...was elevated should be 'were elevated' --please check grammar throughout for minor errors like this one.

Supplemental Table 2: "Significance showed different from former date and was generated by Student's t test" what is the former date? This is not clear.

Supplemental Table 4: Though it may be obvious to the authors, the 12-bp genotype column should be defined in the header. What does "12-bp*2" versus "12-bp" mean? It could easily be interpreted as 2 copies versus one copy?

Line 139: you should state first that DOFCOREZM is the DOF TFBS before just calling it that.

Line 182: pluralize panicle

Line 183: it would help to additionally call C18:3 linoleic acid. More people are familiar with that term.

Line 213: cope should be copy

Lines 222-223: this sentence should be revised to read "of NOG1 haplotype might have originated

in the indica cultivars and was later introduced into japonica cultivars.

Line 249: I suggest you revise this sentence to "Here, we identify a novel gene NOG1 which increases grain number and yield, and show it is a locus that underwent selection during the domestication and diversification process.

Line 266: controlling, not controlled

Line 285: factor that plays

Lines 289-290: I do not think this section of the discussion should end with the finding of repeated motifs increasing expression. As part of the discussion section, it should go one step farther. How can we exploit this knowledge to make better rice? Were repeated TFBS motifs a part of the domestication story?

Line 292: I like this header, but think it should be 'crop breeding' not 'crop production'

Line 300-302: It is essential to rice breeding that we identify novel genes improving yield traits

without any trade-offs for other agronomic or yield-related traits.

Methods: Measurements of endogenous hormones levels. Please change "following published method" to "following a published method"

Reviewer #3 (Remarks to the Author):

This manuscript presented a story about map-based cloning and characterization of NOG1, which could increase grain yield in rice. Further study revealed the difference of NOG1 expression level caused by a 12-bp InDel in the promoter of NOG1 was responsible for the change of grain number. And the authors uncovered the mechanism of NOG1 increasing grain yield working as an enoyl-CoA hydratase (ECH) through beta-oxidation pathway to regulate endogenous JA level. Moreover, NOG1 was subjected to artificial selection during rice domestication. NOG1 is a yield related gene that is very promising to improve grain yield. The whole story sounds good, but the mechanism of NOG1 increasing yield is not clear enough. Some conclusions need to be supported by additional data needed.

1. It is not appropriate to specify NOG1 as a domestication gene. Both NOG1 and *nog1* are detected in cultivars, thus its regulating trait should not be a wild rice specific. The selection of NOG1 is probably occurred in rice genetic improvement for high yield. In addition, they suggested duplication of the 12-bp fragment is occurred after domestication. Therefore NOG1 has nothing to do with domestication.
2. For NOG1 subcellular localization, transient expression of GFP-NOG1 is not reliable, please using other more reliable techniques to test it.
3. The 12-bp insertion carried DOF TFBS, thus they tested Dof12 transcriptional activation to NOG1. Why was Dof12 selected for testing? They should examine the binding of Dof12 to the 12-bp by EMSA or Y1H.
4. It is not correct to specify gain-of function variation of NOG1 haplotype because the two proteins of NOG1 and *nog1* have similar enzymatic activity. Thus the variation of cis-element results expression change and leads to phenotypic variation.
5. Wu et al. reported over-expression of OsDof12 led to smaller panicle resulted from decreased primary and secondary branches number (Wu et al., 2015). And the authors revealed that OsDof12 could increase the expression level of NOG1. But this finding could not explain why NOG1 had a large panicle.
Wu, Q., Li, D., Li, D., Liu, X., Zhao, X., Li, X., . . . Zhu, L. (2015). Overexpression of OsDof12 affects plant architecture in rice (*Oryza sativa* L.). *Frontiers in plant science*, 6.

6. The author revealed that repression of NOG1 could up-regulate the JA level in plants (Fig. 5d). Then JA treatment of Guichao 2 could change the expression of NOG1 (Fig.5 f, j). So

what's the point? Any links to more spikelets per panicle?

7. It is hard to understand the relationship between C18:3 and JA. Line 181-185, the higher content of total fatty acids and unsaturated 18-carbon fatty acids C18:3 in Guichao 2 and the complemented transgenic plants, but the case is completely opposite in Fig 5 c. Line 194, since C18:3 is the synthetic precursor of JA. Does the higher C18:3 content in Guichao 2 and complemented transgenic plants produce low JA content? Line 202-203, decreased the C18:3 content and the endogenous JA levels, thus resulted in increasing the grain number and grain yield in rice. There might be some wrong in the point.

8. More data should be provided to make a stronger conclusion that 12-bp Indel in the promoter was responsible for the expression difference of NOG1. There is some confusion about the results in Fig. 2c and Fig. 4d. In Fig. 2c, the expression of NOG1 in complementary plants is about 3 fold up than SIL176. However, the difference in Fig. 4d is less than 2 fold. Guichao 2 should be added in Fig. 2b-e and Fig. 4e, SIL176 should be added in Fig. 2f-i for better understanding of the comparison.

9. Besides the 12-bp insertion, 15 SNPs were also examined in the promoter of NOG1, how to exclude the possibility of them to regulate the expression of NOG1? Does any SNP damage the cis-elements? It needs to be addressed.

10. The F1 plants (NOG1nog1) between SIL176 and Guichao 2 showed similar performance to Guichao 2 (NOG1NOG1). The function of NOG1 was altered by its expression level, is there no difference in the expression of NOG1 between two genotypes? They should present a figure for expression comparison.

11. In line 168-176, "NOG1 protein has ECH activity, and the ECH activity of the NOG1 protein and nog1 protein are similar..." Why was the activity of the ECH enzyme associated with the expression of NOG1?

12. Fig. 3c, significant marks should be present above the bars to match the statement in line 114. Fig. 3d-f, data of expression difference revealed by in situ should be provided other than the expression pattern. Fig. d, e and f should be separately presented. Line 115, the word "specifically" should be removed or changed to "strongly", since NOG1 expressed ubiquitous almost in any tissue according to results of Fig. 3c.

Reviewer #1's questions and our responses:

R1Q1 (Reviewer #1's question 1): ...Forward and reverse genetics studies conducted in the first part of the article are very thorough and provide solid evidences for the importance *NOG1* for grain production. The identification of *NOG1* is a new and very interesting result for the field of crop improvement...

Our response to R1Q1: Thank you very much for your appreciation.

R1Q2 (Reviewer #1's question 2):

1. The authors indeed detect that JA levels are higher in SIL176 than in Guichao 2 and show that JA application inhibits grain yield. Nevertheless, it remains elusive whether attenuated JA signaling in Guichao 2 is the main cause for the *NOG1*-dependent increases of grain in Guichao 2 is not clear to me. I was notably wondering whether the authors analyzed JA-Ile, the bioactive form in the JA pathway, as well as downstream genes markers for JA signaling to ascertain of an attenuated JA signaling in Guichao 2.

2. β -oxidation is not only important for fatty acid turnover, as also demonstrated in this study by measurements of total fatty acid contents, but also in the last steps of JA synthesis. Peroxisomal β -oxidation is required to convert OPDA into JA. *NOG1* may act more specifically at this stage of the JA pathway. Hence, it would have been interesting to examine OPDA levels to control for a metabolic disruption at stage of the JA pathway.

More generally, it seems to me that comparing JA biosynthetic genes expression between Guichao 2 and SIL176 would have been instrumental to better assess modulations of JA signaling between these two lines.

Our response to R1Q2: Thank you very much for your suggestion.

We have followed your suggestion to investigate the jasmonates contents. The results showed that the levels of jasmonates including jasmonic acid, OPDA and JA-Ile in young panicles of Guichao 2 and the complemented transgenic plants were significantly lower than that in SIL176 and the RNAi transgenic plants (**Fig. 5 d-f**).

Furthermore, we showed that the expression levels of *OsDAD1*, *OsLOX*, *OsAOS2*, *OsAOC* and *OsOPR7* were decreased in Guichao 2 compared with SIL176 (**Supplementary Fig. 7 a-e**). These results were described in the revised manuscript (line 210-217).

R1Q3 (Reviewer #1's question 3): If C18:3 is limiting in Guichao 2 for JA synthesis, does a C18:3 increased JA levels in this line and consequently reduced grain yield to levels observed during JA treatment?

Our response to R1Q3: Thank you for your suggestion. In this study, we focus on the identified a gene, *NOG1*, which regulates grain yield in rice. In addition, we uncovered the mechanism of *NOG1* increasing grain yield working as an ECH enzyme through β -oxidation pathway to regulate endogenous JA level. Despite it has high challenging in establishing C18: 3 on JA synthesis limiting and production regulation, we will further examine the effect of C18:3 in rice in next step of work. Your suggestion is excellent and

pointed out a good research direction for our next work.

R1Q4 (Reviewer #1's question 4): L99 seems to be a too strong statement. *NOG1* is certainly involved, as demonstrated in this study, grain yield; but likely not “the” unique gene influential in this process.

Our response to R1Q4: Thank you for your suggestion. According to your suggestion, the sentence “Taken together, these results demonstrated that LOC_Os01g54860 is **the** *NOG1* gene which **controls** the grain number per panicle and grain yield per plant in rice.” was revised to “Taken together, these results demonstrated that LOC_Os01g54860 is *NOG1* **essential for regulating** the grain number per panicle and grain yield per plant in rice.” in the revised manuscript in line 100-102.

Reviewer #2's questions and our responses:

R2Q1 (Reviewer #2's question 1): ...This finding should be extremely useful to rice breeders and to all plant genetics, that major effect genes underlying traits of interest are still out there to be discovered. I found little to revise in this manuscript other than grammar. The flow is good and the approach is clear and sound...The supplementary tables and figures were also excellent and sufficient, and the statistics and significance tests were appropriate and correct...

Our response to R2Q1: Thank you very much for your appreciation.

R2Q2 (Reviewer #2's question 2): If the editor agrees, (1) it would be opportune to broaden discussion of if there are orthologs in other grains that should be explored for general grain crop improvement, and (2) it would be useful to add a sentence about the earliest evidence of indica being introgressed into japonica (much was the other way around). There are many recent papers on this topic. (3) It would also be useful to add a sentence to the discussion about the role of jasmonic acid response in rice production and in domestication, if there are other changes that have been documented involving this pathway. (4) The abstract could be strengthened to provide more detail about what was found that relates to domestication.

Our response to R2Q2:

Thank you very much for your excellent suggestions.

(1) According to your suggestion, we blasted the *NOG1* amino sequence in NCBI genomic database. We found that the orthologs of *NOG1* exist in various plants such as *Sorghum bicolor*, *Zea mays*, *Setaria italic*, *Triticum aestivum*, and so on. We speculated that the *NOG1* homologous genes may play an important function in different plants. We added these results in the revised manuscript (line 333-337, **Supplementary Fig. 10**).

(2) We have followed your suggestion to add a sentence "Genome sequencing indicated that indica being introgressed into japonica is one kind of domestication models (Huang et al., 2013). Identification of *NOG1* provides a genetic evidence for this model of rice domestication of *indica* and *japonica*." (Please refer to line 249-251 in revised

manuscript).

(3) We have followed your suggestion to add the following sentences in line 218-220 of our revised manuscript: “JA is one of the important signal molecules regulating the development of plants, and higher JA levels might inhibit development of rice spikelet and decrease grain number.”

(4) We have followed your suggestion to add the following sentences in line 25-26 of our revised abstract: “Genetic diversity analysis indicated that *NOG1* has been subjected to directional selection during cultivated rice domestication”.

R2Q3 (Reviewer #2’s question 3): Line 20-21: unclear what “genetically applied” means. Are the authors trying to say genes controlling these traits haven't been pointed to as historic domestication genes?

Our response to R2Q3: We mentioned that “no domestication-related genes were genetically applied to increase grain yield in rice breeding program” in abstract (line 20-21), and the introduction (line 49-50). This sentence clearly stated that, although many domestication-related genes (such as *PROG1*, *SH4 / SHA1*, *LABA1*, *An-1*, *GAD1*) regulate grain yield or yield-related traits, none of those genes has been used in rice breeding program to increase grain yield.

To accommodate your comments, we have made following revision on the sentence (line 20-21; line 50-51): “no domestication-related genes were further utilized to increase grain yield in rice breeding programs.”

R2Q4 (Reviewer #2's question 4):

Line 21: change to "in rice breeding programs"

Lines 30-31: If "Our findings offer new insights into the transition from a lower grain yield to a higher grain yield in rice domestication" then you have to include earlier in the abstract something about evidence that this gene was under selection during the domestication process.

Line 47: "characterized 16-30, only a" should be "characterized 16-30. However, only a"

Our response to R2Q4:

Thank you very much for your excellent suggestion. We have made corrections according to your suggestions.

R2Q5 (Reviewer #2's question 5):

Lines 49-50: (1) again, be more specific about what applications you are referring to, (2) and what a domestication-related gene is. This really requires definition, especially as the community has defined, delimited, and debated domestication genes from diversification genes, and you are introducing a word that embraces both. It's a good word to use but just requires definition, such as giving examples of hallmark domestication genes and the signatures of selection on them from DNA evidence and/or archaeological evidence that let us understand that these caused the changes observed nearly 9000 years ago.

Our response to R2Q5: Thank you very much for your suggestion.

(1) We mentioned that “no domestication-related genes were genetically applied to increase grain yield in rice breeding programs” up to now in abstract (line 20-21), and the introduction (line 49-50). This sentence clearly stated that, although many domestication-related genes (such as *PROG1*, *SH4 / SHA1*, *LABA1*, *An-1*, *GAD1*) regulate grain yield or yield-related traits, none of those genes has been used in rice breeding program to increase grain yield.

To accommodate your comments, we have made following revision on the sentence (line 20-21; line 50-51): “no domestication-related genes were further utilized to increase grain yield in rice breeding programs.”

(2) Generally, the evolution from wild species to cultivated species is divided into two stages (domestication and diversification). "Domestication" occurred in the early evolution stage of crop, which refers to the direct selection from wild ancestral species by the human. So, the domestication characteristics can be used to distinguish crops from their wild ancestry species. The "diversification" occurs after the "domestication", which is a further improvement of the domesticated crops, involving greater improvement in yield, adaptation or quality in crop species (Meyer et al., 2013).

There are two kinds of definition for domestication, both definitions are accepted by scientists in the research area. (i) Broad definition for domestication: the process of domestication included the above-mentioned two stages (“domestication” and “diversification”) (Li et al., 2017). (ii) Narrow definition for domestication: the process of

domestication only included the above-mentioned first stage (“domestication”) excluding the second stage (“diversification”) (Meyer et al., 2013).

In our manuscript, we used broad definition for domestication. Therefore, we stated that *NOG1* is a domestication-related gene. Previous studies during the past 10 years identified several domestication-related genes, including *Sh4/SHA1* (Konishi et al., 2006; Li et al., 2006; Lin et al., 2007), *PROG1* (Jin et al., 2008; Tan et al., 2008), *Bh4* (Zhu et al., 2011), *Rc* (Sweeney et al., 2006), *OsLG1* (Ishii et al., 2013; Zhu et al., 2013), *An-1* (Luo et al., 2013), *LABA1/An-2* (Gu et al., 2015; Hua et al., 2015) and *RAE2/GAD1* (Bessho-Uehara et al., 2016; Jin et al., 2016;). All these genes are considered to be domestication-related genes based on broad definition for domestication, while some of them (such as *Rc*, *Bh4*, *LABA1*) should be diversification-related genes according to narrow definition for domestication.

We are willing to use narrow definition for domestication to define the *NOG1* gene as diversification-related gene if still requested. Thank you!

R2Q6 (Reviewer #2’s question 6):

Line 50: pluralize program

Lines 62: pluralize panicle

Line 73-74: we selected number of grains per main panicle as "the" targeted trait for fine-mapping *NOG1*.

Line 76: change "combining with phenotype" to "using the F₃ generation averaged

phenotypes" (if they are averaged)

Line 79: harbors

Line 84: change 'harbored' to 'contained'.

Line 87: levels...was elevated should be 'were elevated'--please check grammar throughout for minor errors like this one.

Our response to R2Q6:

Thank you very much for your suggestion. We have made correction according to your suggestion.

R2Q7 (Reviewer #2's question 7):

Supplemental Table 2: "Significance showed different from former date and was generated by Student's t test" what is the former date? This is not clear.

Our response to R2Q7: Sorry for the confusion. The sentence was corrected as "Significance showed different from the data in the previous column and was generated by Student's t test".

R2Q8 (Reviewer #2's question 8):

Supplemental Table 4: Though it may be obvious to the authors, the 12-bp genotype column should be defined in the header. What does "12-bp*2" versus "12-bp" mean? It could easily be interpreted as 2 copies versus one copy?

Line 139: you should state first that DOFCOREZM is the DOF TFBS before just calling it

that.

Line 182: pluralize panicle

Line 183: it would help to additionally call C18:3 linolenic acid. More people are familiar with that term.

Line 213: cope should be copy

Lines 222-223: this sentence should be revised to read "of *NOG1* haplotype might have originated in the indica cultivars and was later introduced into japonica cultivars.

Line 249: I suggest you revise this sentence to "Here, we identify a novel gene *NOG1* which increases grain number and yield, and show it is a locus that underwent selection during the domestication and diversification process."

Line 266: controlling, not controlled

Line 285: factor that plays

Our response to R2Q8: Thank you very much for your suggestion. We have made correction according to your suggestion.

R2Q9 (Reviewer #2's question 9):

Lines 289-290: I do not think this section of the discussion should end with the finding of repeated motifs increasing expression. As part of the discussion section, it should go one step farther. (1) How can we exploit this knowledge to make better rice? (2) Were repeated TFBS motifs a part of the domestication story?

Our response to R2Q9: Thank you for your suggestion.

(1) We added the sentences “Overall, we speculate that the repetition of transcription factor binding sites increases gene expression, resulting in increasing grain number and enhancing grain yield. We can increase grain yield by up-regulating the *NOG1* gene expression levels in rice breeding.” in the revised manuscript in line 318-320.

(2) In this study, we compared the diversity of *NOG1* gene in wild rice and cultivated rice, and found that the repeated 12-bp with TFBS motifs was the key loci leading to the functional variation of *NOG1*. Therefore, repeated TFBS motifs should be a part of the whole story of rice domestication.

R2Q10 (Reviewer #2’s question 10):

Line 292: I like this header, but think it should be 'crop breeding' not 'crop production'

Line 300-302: It is essential to rice breeding that we identify novel genes improving yield traits without any trade-offs for other agronomic or yield-related traits.

Methods: Measurements of endogenous hormones levels. Please change "following published method" to "following a published method"

Our response to R2Q10: Thank you very much for your suggestion. We have made correction according to your suggestion.

Reviewer #3’s questions and our responses:

Reviewer #3 raised 12 questions. We have carefully revised the manuscript to fully address all the questions.

R3Q1 (Reviewer #3's question 1): ...*NOG1* is a yield related gene that is very promising to improve grain yield. The whole story sounds good...

Our response to R3Q1: Thank you very much for your appreciation.

R3Q2 (Reviewer #3's question 2): It is not appropriate to specify *NOG1* as a domestication gene. Both *NOG1* and *nog1* are detected in cultivars, thus its regulating trait should not be a wild rice specific. The selection of *NOG1* is probably occurred in rice genetic improvement for high yield. In addition, they suggested duplication of the 12-bp fragment is occurred after domestication. Therefore *NOG1* has nothing to do with domestication.

Our response to R3Q2:

Generally, the evolution from wild species to cultivated species is divided into two stages (domestication and diversification). "Domestication" occurred in the early evolution stage of crop, which refers to the direct selection from wild ancestral species by the human. So, the domestication characteristics can be used to distinguish crops from their wild ancestry species. The "diversification" occurs after the "domestication", which is a further improvement of the domesticated crops, involving greater improvement in yield, adaptation or quality in crop species (Meyer et al., 2013).

There are two kinds of definition for domestication, both definitions are accepted by scientists in the research area. (i) Broad definition for domestication: the process of domestication included the above-mentioned two stages (“domestication” and “diversification”) (Li et al., 2017). (ii) Narrow definition for domestication: the process of domestication only included the above-mentioned first stage (“domestication”) excluding the second stage (“diversification”) (Meyer et al., 2013).

In our manuscript, we used broad definition for domestication. Therefore, we stated that *NOG1* is a domestication-related gene. Previous studies during the past 10 years identified several domestication-related genes, including *Sh4/SHA1* (Konishi et al., 2006; Li et al., 2006; Lin et al., 2007), *PROG1* (Jin et al., 2008; Tan et al., 2008), *Bh4* (Zhu et al., 2011), *Rc* (Sweeney et al., 2006), *OslG1* (Ishii et al., 2013; Zhu et al., 2013), *An-1* (Luo et al., 2013), *LABA1/An-2* (Gu et al., 2015; Hua et al., 2015) and *RAE2/GAD1* (Bessho-Uehara et al., 2016; Jin et al., 2016;). All these genes are considered to be domestication-related genes based on broad definition for domestication, while some of them (such as *Rc*, *Bh4*, *LABA1*) should be diversification-related genes according to narrow definition for domestication.

We are willing to use narrow definition for domestication to define the *NOG1* gene as diversification-related gene if still requested.

R3Q3 (Reviewer #3’s question 3): For *NOG1* subcellular localization, transient expression of GFP-*NOG1* is not reliable, please using other more reliable techniques to test it.

Our response to R3Q3: Thank you very much for your suggestion. According to your suggestion, we transformed *p35S :: NOG1-GFP* fusion vector into rice protoplast cells. Similarly, NOG1-GFP fusion protein was also localized to the nucleus and cytoplasm. We have added this result in the revised manuscript in line 108-112. The result was shown as below:

We constructed a vector harboring *NOG1-GFP* (green fluorescence protein) fusion gene driven by a *CaMV35S* promoter, and then introduced plasmid into onion epidermal cells and rice protoplast cells. The fluorescent signal showed that the NOG1 protein was localized in the nucleus and cytoplasm (**Fig. 3 a-d**).

R3Q4 (Reviewer #3's question 4): The 12-bp insertion carried DOF TFBS, thus they tested Dof12 transcriptional activation to *NOG1*. (1) Why was Dof12 selected for testing? (2) They should examine the binding of Dof12 to the 12-bp by EMSA or Y1H.

Our response to R3Q4:

(1) Previous studies revealed that *OsDof12* was specifically expressed in panicles and related to spike development. We found that mutation of the 12-bp InDel in promoter of the *NOG1* gene significantly reduces the *NOG1* promoter-driven gene expression while such 12-bp InDel serves as the binding site for DOF protein. Therefore, we selected *OsDof12* to test the transient luciferase expression levels in our originally submitted manuscript.

(2) In this study, we focus on the identified a gene, *NOG1*, which regulates grain yield in

rice production. In addition, we further showed that the 12-bp InDel in the upstream of *NOG1* promoter showed the most intense association signal for grain yield in a fine-mapping linkage analysis of 158 cultivars. Promoter assays studies additionally support that transcription factor binding sites within this 12-bp are responsible for the higher expression levels of *NOG1*. We also confirmed that the gene expression of *NOG1* could increase the grain number in rice by transgenic experiment. The interaction of *NOG1* and *OsDof12* is not the responsibility for the integrity of the story, and it will be further examined in the future work. Therefore, in the revised manuscript, we removed the result of *OsDof12*, and the revision will not affect the significance of our study.

R3Q5 (Reviewer #3's question 5): It is not correct to specify gain-of function variation of *NOG1* haplotype because the two proteins of *NOG1* and *nog1* have similar enzymatic activity. Thus the variation of cis-element results expression change and leads to phenotypic variation.

Our response to R3Q5: Thank you for your insightful comments! We have followed your suggestion to change “the gain-of function variation of *NOG1* haplotype” to “12-bp variation in *NOG1* promoter” (line 247 in our revised manuscript).

R3Q6 (Reviewer #3's question 6): Wu et al. reported over-expression of *OsDof12* led to smaller panicle resulted from decreased primary and secondary branches number (Wu et al., 2015). And the authors revealed that *OsDof12* could increase the expression level

of *NOG1*. But this finding could not explain why *NOG1* had a large panicle. Wu, Q., Li, D., Li, D., Liu, X., Zhao, X., Li, X., . . . Zhu, L. (2015). Overexpression of *OsDof12* affects plant architecture in rice (*Oryza sativa* L.). *Frontiers in plant science*, 6.

Our response to R3Q6: In this study, we confirmed that the gene expression of *NOG1* could increase the grain number in rice by transgenic experiment. In addition, the results of transient expression assays and expression investigation of cultivars containing different copy number of the 12-bp indicated that DOF transcription factor binding sites within the 12-bp are responsible for the higher expression levels of Guichao 2 compared to in SIL176. However, Wu et al. found that over-expression of *OsDof12* decreased the primary and secondary branches number, and led to a smaller panicle in Nipponbare. We speculated that *OsDof12* in the rice could affect a series of genes, including some genes which negatively control spike number. Therefore, although *OsDof12* increased the expression of *NOG1*, overexpression of *OsDof12* is still able to reduce the grain number. In addition, in this study, we focus on the identified a gene, *NOG1*, which regulates the rice yield. In the revised manuscript, we removed the result of *OsDof12*, and the revision will not affect the significance of our study.

R3Q7 (Reviewer #3's question 7): The author revealed that expression of *NOG1* could down-regulate the JA level in plants (Fig. 5d). Then JA treatment of Guichao 2 could change the expression of *NOG1* (Fig.5 f, j). So what's the point? Any links to more spikelets per panicle?

Our response to R3Q7: Thank you for your comments. Our results showed that increasing *NOG1* expression levels could down-regulate the endogenous JA levels, and enhance the grain yield in rice (**Fig.5 d**). To further verify whether the JA affects the grain number, we treated Guichao 2 with MeJA at panicles development stage. Consistent with previous observed (Cai et al., 2014; Kim et al., 2009), we also found that higher JA levels decrease grain number.

In addition, we examined the *NOG1* gene expression after JA treatment, and found that JA treatment down-regulates the *NOG1* gene expression. These results are consistent with our conclusion that the *NOG1* gene increases grain number in rice.

R3Q8 (Reviewer #3's question 8): It is hard to understand the relationship between C18:3 and JA. Line 181-185, the higher content of total fatty acids and unsaturated 18-carbon fatty acids C18:3 in Guichao 2 and the complemented transgenic plants, but the case is completely opposite in Fig 5 c. Line 194, since C18:3 is the synthetic precursor of JA. Does the higher C18:3 content in Guichao 2 and complemented transgenic plants produce low JA content? Line 202-203, decreased the C18:3 content and the endogenous JA levels, thus resulted in increasing the grain number and grain yield in rice.

There might be some wrong in the point.

Our response to R3Q8: We are very sorry for the typing errors in our originally submitted manuscript. We changed the sentence "... were higher than..." to "... were lower than..." in our revised manuscript (line 199). Thank you very much!

R3Q9 (Reviewer #3's question 9): (1) More data should be provided to make a stronger conclusion that 12-bp Indel in the promoter was responsible for the expression difference of *NOG1*.

(2) There is some confusion about the results in Fig. 2c and Fig. 4d. In Fig. 2c, the expression of *NOG1* in complementary plants is about 3 fold up than SIL176. However, the difference in Fig. 4d is less than 2 fold.

(3) Guichao 2 should be added in Fig. 2b-e and Fig. 4e, SIL176 should be added in Fig. 2f-i for better understanding of the comparison.

Our response to R3Q9:

(1) We have followed your suggestion to carry out new experiment to verify the 12-bp InDel was responsible for regulating the *NOG1* gene expression levels. 20 cultivars containing two copies of the 12-bp Indel and 20 cultivars containing one copy of the 12-bp Indel were selected to detect their *NOG1* genes expression levels. The results showed that the expression levels of the *NOG1* gene in these varieties containing two copies of the 12-bp were obviously higher than that in varieties containing one copy of the 12-bp. We have showed these results in the revised manuscript in line 151-155.

(2) We have to clarify that the data from Fig. 2c is indeed consistent with Fig. 4d. The expression of *NOG1* in complementary transgenic plants (SIL176 transgenic for the genomic fragment containing *NOG1* promoter and coding region from Guichao 2), were about 3 fold up than SIL176 , consistently, the expression levels of the *NOG1* gene in

protoplasts of Guichao 2 promoter was about 2 fold higher than that of SIL176 promoter (Fig. 4d). These results consistently demonstrate that the *NOG1* gene expression in Guichao 2 was higher in SIL176. It is not surprised that the exact values have slight difference between Fig.2c and Fig 4d since different materials were used.

R3Q10 (Reviewer #3's question 10): Besides the 12-bp insertion, 15 SNPs were also examined in the promoter of *NOG1*, how to exclude the possibility of them to regulate the expression of *NOG1*? Does any SNP damage the cis-elements? It needs to be addressed.

Our response to R3Q10: Thank you very much for your comments. To determine whether the 15 SNPs is responsible for regulating the *NOG1* gene expression, we compared the transient relative activity between two types of *NOG1* promoter (fig4d): (1) the SIL176 promoter containing the 15 SNPs-type 1 (ACTTCTAATCCAGAG) and one copy of the 12-bp, (2) The A3 promoter containing the 15 SNPs-type 2 (TGCCTAGCCTTGTTA) and one copy of the 12-bp. As shown in fig4d, the relative activity of the A3 promoter is similar to that of the SIL176 promoter. These results demonstrate that mutation of the 15 SNPs has no impact on the *NOG1* gene expression, suggesting that 15 SNPs is not responsible for regulating the *NOG1* gene expression.

R3Q11 (Reviewer #3's question 11): The F₁ plants (*NOG1nog1*) between SIL176 and Guichao 2 showed similar performance to Guichao 2 (*NOG1NOG1*). The function of

NOG1 was altered by its expression level, is there no difference in the expression of *NOG1* between two genotypes? They should present a figure for expression comparison.

Our response to R3Q11: Thank you for your suggestion. According to your suggestion, we measured the gene expression of F_1 plants, the results showed that the expression levels of F_1 plants in young panicles were similar with Guichao 2, consistent with phenotype. We added this result in the revised manuscript in line 117-119 and Supplementary Fig. 2 k.

R3Q12 (Reviewer #3's question 12): In line 168-176, "NOG1 protein has ECH activity, and the ECH activity of the NOG1 protein and *nog1* protein are similar..." Why was the activity of the ECH enzyme associated with the expression of *NOG1*?

Our response to R3Q12:

(1) Our in vitro experiment showed that "NOG1 protein has ECH activity, and the ECH activity of the NOG1 protein and *nog1* protein are similar". We found that the equivalent amount of NOG1 and *nog1* protein, which are expressed and purified from yeast, generates the similar ECH enzyme activity (Fig. 5a).

(2) Because the expression levels of *NOG1* in Guichao 2 were higher than that in SIL176, "the levels of the ECH enzyme" but not "the activity of the ECH enzyme" in Guichao 2 is expectedly higher than that in SIL176. Thank you for your insightful comments! We have made revision according to your suggestions.

R3Q13 (Reviewer #3's question 13): (1) Fig. 3c, significant marks should be present above the bars to match the statement in line 114. (2) Fig. 3d-f, data of expression difference revealed by in situ should be provided other than the expression pattern. (3) Fig. d, e and f should be separately presented. (4) Line 115, the word “specifically” should be removed or changed to “strongly”, since *NOG1* expressed ubiquitous almost in any tissue according to results of Fig. 3c.

Our response to R3Q13: Thank you very much for your suggestions.

(1) We have made revision according to your suggestions (significant marks are present in fig 3).

(2) We have followed your suggestion to add the RNA *in situ* hybridization of SIL176.

These new data consistently demonstrated that the *NOG1* transcript levels

(hybridization signals) in SIL176 were significantly weakened compared with Guichao 2.

We added this result in revised manuscript in line 119-121.

(3) We have made revision according to your suggestions (Fig. 3 d-f is now separately presented).

(4) Line 115, the word “specifically” is removed.

Thank you!

Reviewer Comments:

Reviewer #1 (Remarks to the Author):

This revised manuscript is significantly improved compared to its earlier version. Notably, authors provide additional jasmonate profiling results as well data about transcript levels for jasmonate biosynthetic genes. All these data and the fact that they support a reduction of jasmonate signalling in Guichao2, are in line with the hypothesis previously formulated by authors. The manuscript still lacks a clear mechanistic understanding of the exact role of NOG1. But, this role, most particularly in regulating C18:3 availability in panicles for JA biosynthesis, can be addressed in future work.

Reviewer #2 (Remarks to the Author):

I appreciate the work the authors have done to address my initial concerns, and the additional experiments they did to address excellent points made by other reviewers.

I have only minor edits to request.

 "and provide a novel favorable gene to increase rice production" -- it's strange to say novel when it was a diversification gene-- perhaps better to just say a favorable gene

Negative Tajima's D is not directional selection. It means there has been population expansion after a recent sweep. The authors should carefully consider if they want to still say directional selection.

Line 37: add the latin binomial for wild rice, *Oryza rufipogon* Griff. -- otherwise you don't define it.

"In recent years, several genes controlling yield-related traits have been
48 characterized 16-30. However, only a few genes can actually increase yield and can be used
49 in breeding programs due to the dynamic balance among these three main components
50 of rice grain yield. In particular, no domestication-related genes were further utilized to
51 increase grain yield in rice breeding programs." --This sounds awkward-- maybe it is not
coming through in translation. How do you know there are only a few genes controlling this
trait? Do you mean only a few genes found to play a role in this trait so far? Perhaps don't say
"were further utilized" here. It sounds like your experiment. I think you mean to say 'are
currently being targeted in crop yield improvement programs'

And even so, then you need to state how you screened for that across the world. Is that based on
a literature search? Maybe make the statement softer.

There are several small spelling errors, such as line 175 14th exons should be exon, line 296
should be controlled not controlling. There need to be a lot of minor edits made so the English
sounds better if this is to be published in Nature Communications.

The new writing has a lot of repetitive sentences that could be tightened to make the manuscript
flow.

Lines 223-226: Further we examined the NOG1 gene expression after JA treatment and found
that the NOG1 gene expression in Guichao2 was down-regulated by JA treatment (Fig. 5 g-j).
These results demonstrated that the JA treatment down-regulated NOG1 gene expression and
decreased grain number in rice.

Line 234: where is Tajima's D reported? Why not report the actual value not just the P value
here? Show the value.

Line 232-238: It is really hard to interpret the pattern of nucleotide diversity when only such a
small region is plotted. As cultivated rice has a substantially reduced diversity, I don't see this as
compelling evidence that there has been a sweep.

Line 232: The authors state "The results showed that the nucleotide diversity of the region
surrounding 12-bp InDel in cultivated rice was significantly lower than that in wild rice

(Supplementary Fig. 8)." Could the authors mark on this plot where the position of the 12bp indel is and other SNPs? Also, for Suppl Fig 3a, could the authors please note where the TSS is?

Perhaps the authors should point this out and a single copy fixed in wild rice (correct?) might be adaptive in the wild.

The authors reference #38 and say their result of low nucleotide diversity is consistent with sweeps found. This is vague. Did reference 38 find a sweep in the very location of the NOG1 promoter? If so, say that. Is there a sweep around this locus? There are several genome-wide screens published to examine for this. Or is this typical reduced nucleotide diversity across the genome as a result of domestication that we are observing.

Lines 249-251. This new wording does not make sense.

"Genome sequencing indicated indica being introgressed into japonica is one kind of domestication models³⁸. Identification of NOG1 provides a genetic evidence for this model of rice domestication of indica and japonica."

The authors should stay away from discussion of domestication "models", especially when they make no literature reference to these models. They only need to make the point that gene flow between japonica and indica occurred in both directions throughout rices domestication history. It would be much clearer for them to just say this and follow it with the suggestion that the 2-copy 12-bp insert may be a case of introgression from indica into japonica (they wrote the second half of this point clearly in line 248).

OPDA and JA-Ile must be named in full in the ms main text and now just as abbreviations.

277 Here, we identify a novel gene NOG1 which increases grain number and yield, and show

278 it is a locus that underwent selection during the domestication and diversification

279 process. -- the authors don't show it underwent selection during domestication AND diversification. This is confusing. Best to leave the definition broad and state something to the effect of 'over the course of rice diversification, perhaps originating in Indica rice'

Reviewer #3 (Remarks to the Author):

. The authors well answered or addressed most of the questions I raised. But the explanation for the most important question on domestication is not satisfied. I suggest the authors describe the story on the basis of narrow definition of domestication. Like reviewer 2 complained, I do not like the broaden definition on domestication. I think most readers would be like to accept the narrow definition, especially when mention wild rice. After domestication, artificial selection would be occurred for pursuing high yield production. NOG1 should be suffered selection in the process of genetic improvement. Genetic improvement should not be considered into domestication. Otherwise the modern breeders are still working in domestication. Obviously it is not correct.

. As to the point that no domestication-related genes were further utilized to increase grain yield in rice breeding programs. I do not agree with this point because all the domesticated traits were existed in cultivars and favorable for high yield production. Like shattering genes, the favorable alleles were carried by all modern cultivars that increase yield via decreasing loss-of-yield, which means all domesticated genes are always utilized by human.

. An elite indica variety Guichao 2 is not correct. Please change to a high yielding indica variety

. Supplementary table 1 add additive effects, positive value means Guichao 2 carried alleles increasing trait values. Dominance effect of alleles changes to dominance effect.

. Supplementary fig 1: genotype change to “genome constitution”.

. “Genetic linkage analysis within 6230 F2 individuals derived from the cross between Guichao 2 and SIL176 using the F3 generation phenotypes revealed that NOG1 was delimited to a 4.3-kb region” change to “progeny test of 6230 F2 individuals derived from the cross between Guichao 2 and SIL176 revealed”. Please polish the language writing through the MS.

. L249-250, Reference 38 did not claim indica being introgressed into japonica. In fact they pointed that indica is developed from the cross between japonica and wild rice and japonica was earlier developed than indica.

Reviewer #1 (Remarks to the Author):

This revised manuscript is significantly improved compared to its earlier version. Notably, authors provide additional jasmonate profiling results as well data about transcript levels for jasmonate biosynthetic genes. All these data and the fact that they support a reduction of jasmonate signaling in Guichao 2, are in line with the hypothesis previously formulated by authors. The manuscript still lacks a clear mechanistic understanding of the exact role of *NOG1*. But, this role, most particularly in regulating C18:3 availability in panicles for JA biosynthesis, can be addressed in future work.

Our response to Reviewer #1:

Thank you very much for your appreciation.

Reviewer #2 (Remarks to the Author):

R2Q1 (Reviewer #2's question 1):

I appreciate the work the authors have done to address my initial concerns, and the additional experiments they did to address excellent points made by other reviewers. I have only minor edits to request.

Our response to R2Q1:

Thank you very much for your appreciation.

R2Q2 (Reviewer #2's question 2):

(1)"and provide a novel favorable gene to increase rice production" -- it's

strange to say novel when it was a diversification gene-- perhaps better to just say a favorable gene

(2) Negative Tajima's D is not directional selection. It means there has been population expansion after a recent sweep. The authors should carefully consider if they want to still say directional selection.

(3) Line 37: add the latin binomial for wild rice, *Oryza rufipogon* Griff. -- otherwise you don't define it.

Our response to R2Q2:

Thank you very much for your suggestions.

(1) We have made revision according to your suggestion, removed the "novel" in revised manuscript (line 34).

(2) We removed the claims that the *NOG1* has undergone directional selection.

(3) We added the latin binomial for wild rice in line 40. Thank you!

R2Q3 (Reviewer #2's question 3):

"In recent years, several genes controlling yield-related traits have been characterized. However, only a few genes can actually increase yield and can be used in breeding programs due to the dynamic balance among these three main components of rice grain yield. In particular, no domestication-related genes were further utilized to increase grain yield in rice breeding programs."

--This sounds awkward-- maybe it is not coming through in translation. How do you know there are only a few genes controlling this trait? Do you mean only a

few genes found to play a role in this trait so far? Perhaps don't say "were further utilized" here. It sounds like your experiment. I think you mean to say 'are currently being targeted in crop yield improvement programs'. And even so, then you need to state how you screened for that across the world. Is that based on a literature search? Maybe make the statement softer.

Our response to R2Q3:

Thank you very much for your suggestion! According to your suggestion, we change the sentences "In recent years, several genes controlling yield-related traits have been characterized ... were further utilized to increase grain yield in rice breeding programs." to "In recent years, several genes controlling yield-related traits have been characterized. However, the molecular mechanisms increasing grain yield during rice improvement are still largely unknown." (line 50-56)

R2Q4 (Reviewer #2's question 4):

(1) There are several small spelling errors, such as line 175 14th exons should be exon, line 296 should be controlled not controlling. There need to be a lot of minor edits made so the English sounds better if this is to be published in Nature Communications.

(2) The new writing has a lot of repetitive sentences that could be tightened to make the manuscript flow.

Lines 223-226: Further we examined the *NOG1* gene expression after JA

treatment and found that the *NOG1* gene expression in Guichao2 was down-regulated by JA treatment (Fig. 5 g-j). These results demonstrated that the JA treatment down-regulated *NOG1* gene expression and decreased grain number in rice.

Our response to R2Q4:

(1) We have made revision according to your suggestions.

(2) The sentences are shortened to "Further examination showed that the *NOG1* gene expression in Guichao2 was down-regulated by JA treatment (**Fig. 5 g-j**). These results demonstrated that the JA treatment down-regulated *NOG1* gene expression and decreased grain number in rice." Thank you!

R2Q5 (Reviewer #2's question 5):

Line 234: where is Tajima's D reported? Why not report the actual value not just the P value here? Show the value.

Line 232-238: It is really hard to interpret the pattern of nucleotide diversity when only such a small region is plotted. As cultivated rice has a substantially reduced diversity, I don't see this as compelling evidence that there has been a sweep.

Line 232: The authors state "The results showed that the nucleotide diversity of the region surrounding 12-bp InDel in cultivated rice was significantly lower than that in wild rice (Supplementary Fig. 8)." Could the authors mark on this plot where the position of the 12bp indel is and other SNPs?

The authors reference #38 and say their result of low nucleotide diversity is consistent with sweeps found. This is vague. Did reference 38 find a sweep in the very location of the *NOG1* promoter? If so, say that. Is there a sweep around this locus? There are several genome-wide screens published to examine for this. Or is this typical reduced nucleotide diversity across the genome as a result of domestication that we are observing.

Our response to R2Q5:

Thank you very much for your excellent suggestions! According to your suggestion, based on the difficulty in inferring selection by nucleotide diversity in rice, we removed the claims that the *NOG1* has undergone directional selection, and we agree with you that it would be potentially misleading to refer to *NOG1* as a domestication-related gene. At the same time, we stay away from discussion of domestication "models" using nucleotide diversity analysis, especially when they make no literature reference to these models. Instead, we focus the gene flow between *japonica* and *indica* occurred in both directions throughout rice domestication history, and just suggested that the two copies of the 12-bp insertion may be a case of introgression from *indica* into *japonica*. Overall, we removed the claims that the *NOG1* has undergone directional selection according to your suggestions, and deleted some of the nucleotide diversity results in the revised manuscript (line 26-28, 245-271). We believe that will make our results more concise and clear, and the revision will not affect the significance of our study.

R2Q6 (Reviewer #2's question 6):

- (1) For Suppl Fig 3a, could the authors please note where the TSS is?
- (2) Perhaps the authors should point this out and a single copy fixed in wild rice (correct?) might be adaptive in the wild.

Our response to R2Q6:

Thank you very much for your excellent suggestions.

- (1) We noted where the TSS, the 12-bp InDel and 15 SNPs in Supplementary Fig. 3a.
- (2) We added the sentence "The single copy of the 12-bp fixed in wild rice might be adaptive in the wild" in the revised manuscript in line 258-259.

R2Q7 (Reviewer #2's question 7):

Lines 249-251. This new wording does not make sense.

"Genome sequencing *indicated indica* being introgressed into *japonica* is one kind of domestication models. Identification of *NOG1* provides a genetic evidence for this model of rice domestication of *indica* and *japonica*."

The authors should stay away from discussion of domestication "models", especially when they make no literature reference to these models. They only need to make the point that gene flow between *japonica* and *indica* occurred in both directions throughout rice domestication history. It would be much clearer for them to just say this and follow it with the suggestion that the 2-copy 12-bp insert may be a case of introgression from *indica* into *japonica* (they wrote the second half of this point clearly in line 248).

Our response to R2Q7:

Thank you for your suggestion. According to your suggestion, we did not discuss the domestication models, and removed the sentences "Genome sequencing indicated that *indica* being introgressed into *japonica* is one kind of domestication models. Identification of *NOG1* provides a genetic evidence for this model of rice domestication of *indica* and *japonica*."

R2Q8 (Reviewer #2's question 8):

(1) OPDA and JA-Ile must be named in full in the main text and now just as abbreviations.

(2) Here, we identify a novel gene *NOG1* which increases grain number and yield, and show it is a locus that underwent selection during the domestication and diversification process. -- the authors don't show it underwent selection during domestication AND diversification. This is confusing. Best to leave the definition broad and state something to the effect of 'over the course of rice diversification, perhaps originating in *indica* rice'

Our response to R2Q8: Thank you for your suggestion.

(1) We have made revision according to your suggestions.

(2) We have followed your suggestion to remove the sentence "... and show it is a locus that underwent selection during the domestication and diversification process."

Reviewer #3 (Remarks to the Author):

R3Q1 (Reviewer #3's question 1):

The authors well answered or addressed most of the questions I raised. But the explanation for the most important question on domestication is not satisfied. I suggest the authors describe the story on the basis of narrow definition of domestication. Like reviewer 2 complained, I do not like the broaden definition on domestication. I think most readers would be like to accept the narrow definition, especially when mention wild rice. After domestication, artificial selection would be occurred for pursuing high yield production. *NOG1* should be suffered selection in the process of genetic improvement. Genetic improvement should not be considered into domestication. Otherwise the modern breeders are still working in domestication. Obviously it is not correct.

Our response to R3Q1:

Thank you for your excellent suggestion. We have considered your suggestion very carefully. We agree with you and the reviewer 2 that it would be potentially misleading to refer to *NOG1* as a domestication gene. We did not insist that *NOG1* is a domestication-related gene, and made the corrections to reflect this point in the revised manuscript (line 19-21, 33-34 and 51-55)

R3Q2 (Reviewer #3's question 2):

As to the point that no domestication-related genes were further utilized to increase grain yield in rice breeding programs. I do not agree with this point

because all the domesticated traits were existed in cultivars and favorable for high yield production. Like shattering genes, the favorable alleles were carried by all modern cultivars that increase yield via decreasing loss-of-yield, which means all domesticated genes are always utilized by human.

Our response to R3Q2:

Thank you for your suggestion. This sentence can cause ambiguity, and may be confusing. While corrected the views on domestication, we removed this sentence in the revised manuscript.

R3Q3 (Reviewer #3's question 3):

(1) An elite *indica* variety Guichao 2 is not correct. Please change to a high yielding *indica* variety

(2) Supplementary table 1 add additive effects, positive value means Guichao 2 carried alleles increasing trait values. Dominance effect of alleles changes to dominance effect.

(3) Supplementary fig 1: genotype change to "genome constitution".

(4) "Genetic linkage analysis within 6230 F2 individuals derived from the cross between Guichao 2 and SIL176 using the F3 generation phenotypes revealed that *NOG1* was delimited to a 4.3-kb region" change to "progeny test of 6230 F2 individuals derived from the cross between Guichao 2 and SIL176 revealed".

Please polish the language writing through the MS.

(5) L249-250, Reference 38 did not claim *indica* being introgressed into *japonica*.

In fact they pointed that *indica* is developed from the cross between *japonica* and wild rice and *japonica* was earlier developed than *indica*.

Our response to R3Q3:

Thank you very much for your excellent suggestions. We have made corrections according to your suggestions.

Thank you!